# Regulatory variation controlling architectural pleiotropy in maize

Edoardo Bertolini [1], Brian R. Rice[2], Max Braud [1], Jiani Yang [1], Sarah Hake [3,4], Josh Strable [5], Alexander E. Lipka [2] & Andrea L. Eveland [1] ✉

An early event in plant organogenesis is establishment of a boundary between the stem cell containing meristem and differentiating lateral organ. In maize (*Zea mays*), evidence suggests a common gene network functions at boundaries of distinct organs and contributes to pleiotropy between leaf angle and tassel branch number, two agronomic traits. To uncover regulatory variation at the nexus of these two traits, we use regulatory network topologies derived from specific developmental contexts to guide multivariate genome-wide association analyses. In addition to defining network plasticity around core pleiotropic loci, we identify new transcription factors that contribute to phenotypic variation in canopy architecture, and structural variation that contributes to *cis*-regulatory control of pleiotropy between tassel branching and leaf angle across maize diversity. Results demonstrate the power of informing statistical genetics with context-specific developmental networks to pinpoint pleiotropic loci and their *cis*-regulatory components, which can be used to fine-tune plant architecture for crop improvement.

Plant architecture has been an important target of selection in crop domestication and improvement[1]. The domestication of maize from its wild progenitor teosinte involved major architectural shifts in apical dominance and ear morphology that were pivotal in generating the backbone of a highly productive crop[2]. Improvements to maize architecture through breeding over the last century have substantially contributed to exponential yield gains; more compact plants with upright leaves and smaller, fewer, or upright tassel branches enable increased planting densities while enhancing photosynthetic efficiency in the lower canopy[3–6]. These architectural traits are outputs of endogenous developmental programs, intricately connected through gene networks that we are just beginning to unravel.

Variation in plant architecture typically involves changes in the placement, number, or orientation of lateral organs, which are initiated from populations of pluripotent stem cells called meristems. Often, this is regulated by meristem determinacy

programs[7], and the concept of signaling centers acting in boundaries adjacent to meristems was proposed to modulate meristem determinacy and architectural diversity[8]. During organogenesis, a boundary domain is established between the meristem and differentiating lateral organs to restrict meristem maintenance and organ identity genes to their respective zones[9]. Leaf angle (LA) in grasses is largely determined by patterning and growth of the ligule and auricles, structures that characteristically define the blade-sheath boundary and together act as a hinge to allow the leaf to recline and absorb sunlight[9].

Several genes expressed at initiating ligules are also expressed at the boundaries of other lateral organs, such as developing leaf primordia and tassel branches[10,11]. Among these are *liguleless* (*lg*)*1* and *2*, which encode squamosa binding protein (SBP) and bZIP transcription factor (TF)s, respectively. Loss-of-function mutants in these genes have compromised ligule and auricle development, resulting in more upright leaves[12,13], but also defects in tassel branch number (TBN) and

[1]Donald Danforth Plant Science Center, St. Louis, MO 63132, USA. [2]Department of Crop Sciences, University of Illinois, Urbana-, Champaign, IL 61801, USA. [3]Plant Gene Expression Center, USDA-ARS, Albany, CA 94710, USA. [4]Plant and Microbial Biology Department, University of California, Berkeley, CA 94720, USA. [5]Department of Molecular and Structural Biochemistry, North Carolina State University, Raleigh, NC 27695, USA. ✉e-mail: aeveland@danforthcenter.org

angle (TBA) (Supplementary Note 1). Mutants in *lg2* make few to no tassel branches that are upright compared to those of normal siblings[14], and *lg1* has significantly fewer branches, but the phenotype is less severe than in *lg2*[15]. Several other maize mutants also show pleiotropic defects between leaf and tassel architecture traits, including those with altered brassinosteroid (BR) signaling, a plant growth hormone that modulates cell division and expansion[16,17] (Supplementary Note 1).

Core gene regulatory modules appear to underlie the formation of a boundary, whether it is the boundary at the ligule, the boundary between leaf primordium and meristem, or between the tassel branch and rachis. Similar to seminal findings in animal systems[18], these common modules have likely been co-opted for the development of distinct tissues and underlie pleiotropy found between LA and TBN, important agronomic traits in maize improvement. Although genome-wide association studies (GWAS) identified significant SNP-trait associations for TBN in proximity to *lg1* and *lg2*, pleiotropy between these traits is less prominent in natural populations[19]. This is likely due to regulatory variation within natural diversity, e.g., *cis*-regulatory elements that specify spatiotemporal patterning of gene expression and are hypothesized to be key drivers of phenotypic variation[20,21].

Pleiotropy, the effect of a gene on multiple phenotypic characters, is a significant cause of evolutionary constraint, and regulatory variation in pleiotropic loci underpins adaptive evolution and developmental plasticity[22–24]. Looking forward, the success of new technologies that allow precise engineering of genomes and pathways will depend on our understanding of pleiotropy in gene networks and devising ways of dissociating pleiotropic effects during crop improvement[25,26]. The pleiotropy that exists between LA and TBN in maize, and the mutants that provide a genetic framework for linking these traits, make it an ideal system for dissecting control points in context-specific gene regulation.

Here, we leverage this system along with a novel approach that integrates developmental biology, network graph theory, and quantitative genetics to identify new factors and regulatory variations contributing to pleiotropy in tassel and leaf architecture. We demonstrate that strategic integration of developmental context-specific biological data to inform reduced marker sets in association studies can enable the discovery of pleiotropic loci of small effect size, which are more agronomically relevant and typically masked by large effect loci.

## Results

### A transcriptional framework for molecular explorations of tassel branching and leaf angle

To delineate the gene networks underlying tassel branching and ligule development, and tissue-specific rewiring around pleiotropic loci, we leveraged a panel of maize mutants with developmental defects in TBN, LA or both, all well-introgressed into the B73 genetic background (Fig. 1a). Mutants with altered TBN and/or TBA are described in the Supplementary Note 1 and include: *lg1*, *lg2*, *wavy auricle on blade1* (*wab1*) and the dominant *Wab1-R* allele, *ramosa1* (*ra1*), *ra2*, *brassinosteroid insensitive1* (*bri1*)-RNAi and *bin2-RNAi* (Supplementary Note 1). These genetic stocks, including B73 control plants, were grown in environmentally controlled chambers to enable precise developmental staging of different genotypes. Immature tassel primordia were hand-dissected immediately before and after primary branch initiation (stage 1 and 2, respectively; Supplementary Fig. 1). We also sampled across a five-stage developmental trajectory from normal B73 tassels; the additional three stages representing different meristem transitions after primary branch initiation. Mutants with defects in LA included *lg1*, *lg2*, *Wab1-R*, *bri1-RNAi*, *bin2-RNAi*, and *feminized upright narrow* (*fun*), which makes a ligule but no auricles (Supplementary Note 1). We sampled two sections of vegetative shoot from these mutants and B73 controls, which we refer to as shoot apex 1 and 2: shoot apex 1 includes the shoot apical meristem (SAM) and cells that are pre-patterned to be

ligule; and shoot apex 2 was taken above the meristem to include leaf primordia with developed ligules (Supplementary Fig. 1).

RNA-seq was used to profile gene expression across 140 samples (Supplementary Data 1). Principal component analysis (PCA) of the normal B73 samples showed clear separation by tissue type and mutant samples cleanly separated by genotype and tissue, indicating distinct transcriptome profiles (Supplementary Fig. 2). Samples across the five stages of B73 tassel development plotted in PCA space revealed a continuous transcriptional gradient reflecting progression through specific axillary meristem types and progression from indeterminate to determinate states. We first examined early shifts in gene expression during primary branch initiation and leaf differentiation in normal B73 plants, which included genes related to meristem identity and determinacy, organ specification, and growth hormone activity (Fig. 1b and Supplementary Fig. 3). We tested for enrichment of functional categories using Gene Ontologies (GO) within differentially expressed (DE) gene sets that were either up- or down-regulated during the shift from meristem to organ differentiation, i.e., between stage 1 and 2 tassels (1719) and shoot apex 1 and 2 (1858) (Fig. 1b and Supplementary Data 2, 3). As expected, the enrichment of several GO categories shifted in common during tassel branch and ligule development, e.g., those related to meristem activity and determinacy and inflorescence morphogenesis, consistent with common sets of genes recruited for boundary establishment and organ development. GO terms related to leaf morphogenesis showed opposite expression between tissues, consistent with the leaf development program being suppressed during tassel branch outgrowth (Fig. 1b). Several plant hormone pathways were overrepresented and tended to show different trends during tassel branch and ligule development. For example, auxin and BR genes showed common expression trajectories in the two developmental programs, whereas jasmonic acid (JA), salicylic acid (SA), and particularly gibberellic acid (GA), showed opposite trends (Fig. 1b). Members of certain developmental TF families showed specific patterns of overrepresentation in tassel branching and leaf development (Supplementary Data 4, 5). For example, ZINC FINGER HOMEODOMAIN (ZHD) and GRAS family TFs were overrepresented during ligule development, several of which were annotated with GO terms related to GA signaling (Supplementary Fig. 4).

### Heterochronic shifts in gene expression underlie tassel mutant phenotypes

Along the defined expression trajectory of the B73 tassel, we interpolated the mutant tassel expression data to capture deviations from this trajectory, i.e., shifts in heterochrony. We fit a smooth spline regression model using expression values of the 500 most dynamically expressed genes across normal tassel development and used this model to classify samples relative to their transcriptomes, measured in expression time (ET) units (Fig. 1c). Expression profiles of known marker genes in maize inflorescence development (e.g., meristem identity genes) strongly support the ET classifications (Supplementary Fig. 5). Mutant expression data showed clear shifts from normal tassel branch initiation and development, although phenotypic differences were largely not observed at these stages (Fig. 1c). For example, ET classifications of BR signaling mutants, *bri1*-RNAi and *bin2*-RNAi, were notably shifted compared to controls at stage 1 but not stage 2. One interpretation is that transcriptional events modulated by BR signaling during tassel development occur early at branch initiation and become more similar to controls during branch outgrowth. The *bri1* and *bin2* genes encode positive and negative regulators of BR signaling, respectively, and the RNAi mutants display opposite phenotypes for LA and TBA (Fig. 1a and Supplementary Note 1). BRs play an important role in maintaining boundary domain identity through control of cell division[27,28], which is consistent with strong shifts in BR mutant gene expression as boundaries are developed during primary branch initiation.

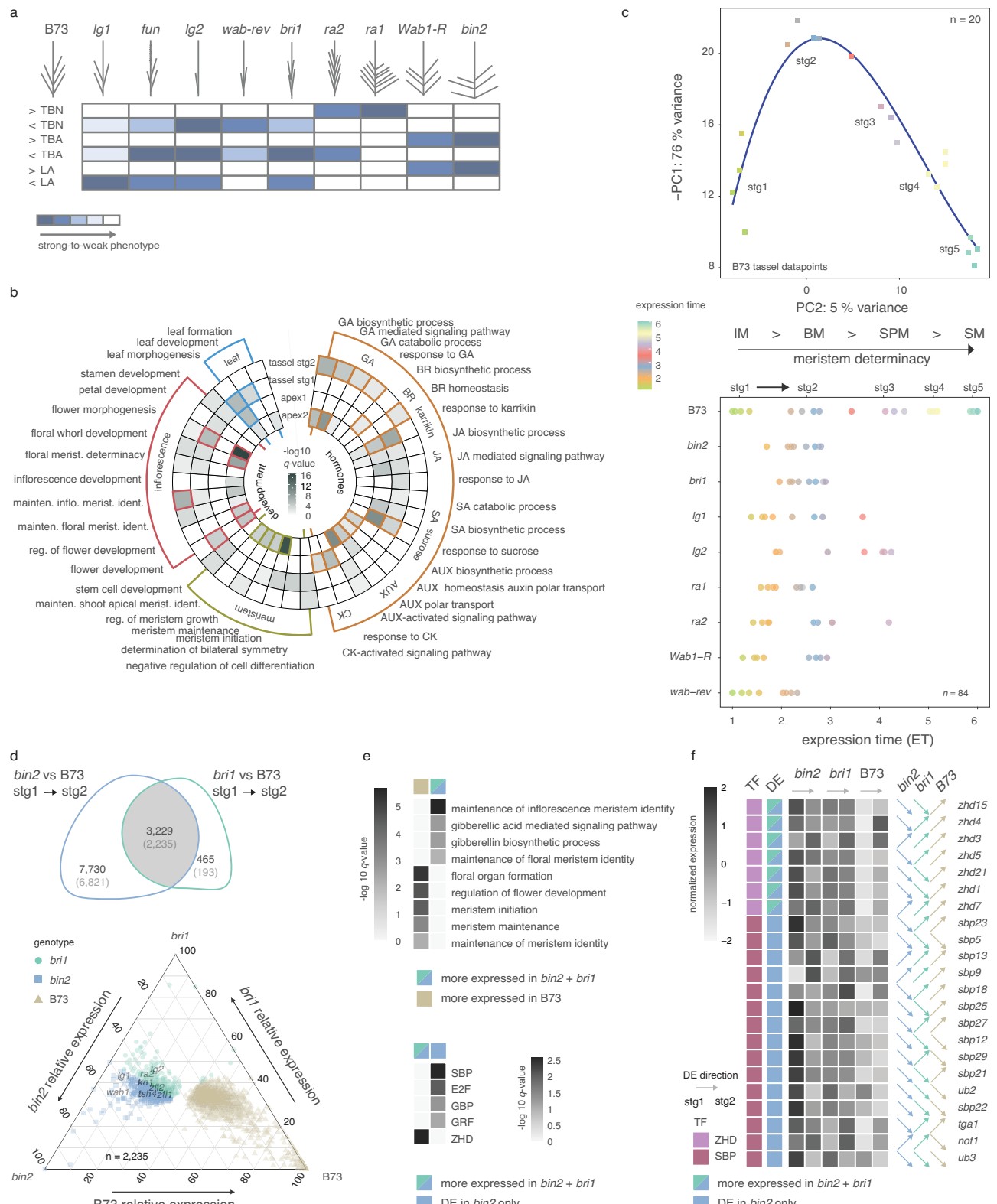

Shifts in gene expression between tassel primordia at stages 1 and 2 were compared between normal B73 controls and the two BR signaling mutants. Of 3229 DE genes that showed a differential expression trajectory in both mutants compared to controls, 2235 showed a larger difference at stage 1 (Fig. 1d and Supplementary Fig. 6). These latter DE genes were enriched for functions related to both meristem maintenance and identity, consistent with important boundary functions, and GA biosynthesis and signaling, suggesting cross-talk between BR

and GA pathways (Fig. 1e). Developmentally dynamic genes that were expressed higher in BR signaling mutants at stage 1 were also significantly enriched for ZHD TF family members ($q = 2.45e^{-03}$) (Fig. 1f and Supplementary Data 6–8). Notably, we observed significant enrichment ($q = 1.87e^{-02}$) of SBP TFs among genes up-regulated in *bin2*-RNAi mutants only during primary branch initiation when compared to normal plants, several of which have been implicated in grass inflorescence development (Fig. 1f)[29,30].

**Fig. 1 | Transcriptional analyses across maize mutants with defects in leaf angle and tassel branching. a** Panel of maize mutants used in this study depicts tassel branching phenotypes compared to B73 controls. Heatmap color scale indicates the severity of tassel branch number (TBN), tassel branch angle (TBA), and leaf angle (LA) phenotype deviations from B73. **b** Enrichment of GO terms associated with DE gene sets during shifts from stage (stg)1 to stg2 tassels and from shoot apex 1 to shoot apex 2. **c** A two-dimensional section of first and second PCs from the B73 tassel developmental gradient ($n = 20$). The blue line represents the b-spline fit with three degrees of freedom modeled within the dataset and classifies samples relative to developmental progression coded as expression time (ET). Colored squares represent the 20 tassel samples in ET. The dot plot below represents stg1 and 2 tassel primordia from mutants with data points color-coded relative to B73 ET. Stages of meristem development along normal tassel developmental time are indicated: inflorescence meristem (IM), branch meristem (BM), spikelet pair meristem (SPM), and spikelet meristem (SM).

**d** Venn diagram shows genes DE in tassels during the shift from stg1 to 2 and DE in *bri1-RNAi* and *bin2-RNAi* mutants compared to B73 controls. Those with stronger deviations at stg1 are in parentheses. The ternary plot below shows the relative expression of genes commonly mis-expressed between the two mutants compared to B73 (stg1). Each dot represents a gene, and its coordinates indicate relative expression among genotypes. Some classical maize genes are noted. **e** The top heatmap shows GO terms enriched among DE genes commonly expressed (higher or lower) in BR mutants compared to B73 in stg1 tassels (from the ternary plot in d); the bottom heatmap shows overrepresented TF classes among genes expressed higher in both BR mutants at stg1 and those expressed higher in *bin2* only compared to B73. **f** Relative gene expression trends for ZHD and SBP family members expressed higher in *bin2* and/or *bri1* mutants compared to B73 at stg1. Arrows indicate the direction of expression change between stg1 and 2 tassels. Source data are provided as a Source Data file.

## Gene network plasticity around pleiotropic loci in different developmental contexts

To determine gene regulatory network interactions during tassel branch and ligule development we integrated the expression data into "tassel" and "leaf" gene co-expression networks (GCNs) and gene regulatory networks (GRNs). A common set of 22,000 expressed genes was used to generate the two GCNs by weighted gene co-expression network analysis (WGCNA)[31]. Genes were grouped based on their similar expression patterns into 16 and 18 co-expressed modules in "tassel" and "leaf" networks, respectively, which are indicated by color (Supplementary Fig. 7 and Supplementary Data 9, 10).

To assess the extent of module conservation between the two GCNs, we conducted a co-expression module preservation analysis based on a permutation method ($n = 1000$; see Methods). We found that 11,221 nodes (52% of the commonly expressed genes) were statistically preserved ($\alpha = 0.05$, Fisher's exact test) and that 69% of the "tassel" modules were conserved with one or more sub-modules in the "leaf" GCN (Fig. 2a and Supplementary Data 11). Two "tassel" GCN modules ("brown" and "blue") were the most preserved, with more than 80% module conservation. The "tassel" "brown" module included overrepresented genes associated with the regulation of meristem growth (GO:0010075, $q = 7.74e^{-05}$) and maintenance of shoot apical meristem identity (GO:0010492, $q = 0.0035$) (Supplementary Data 12). GO enrichment analysis was performed on conserved "tassel-leaf" GCN sub-modules (Supplementary Data 13). For example, the "tassel" "brown" GCN module included two conserved sub-modules in the "leaf" GCN; i.e., a "leaf" "green" sub-module (nodes = 176, Fisher's exact test $P = 5.83e^{-13}$) and a "leaf" "red" sub-module (nodes = 591, Fisher's exact test $P = 0$) that were enriched for genes involved in GA (GO:0009740, $q = 5.85e^{-05}$) and BR (GO:0009742, $q = 0.03$) mediated signaling pathways, respectively.

To visualize network rewiring within preserved sub-modules, we conducted a complementary GRN analysis to infer edge directionality. By overlaying the network connections from the GCN and GRN, we observed that several highly connected regulatory TFs had conserved network connections in "tassel" and 'leaf' networks. For example, ZHD15 (*Zm00001d003645*) showed a high degree of edge conservation in the "brown-green" sub-module, suggesting it is a conserved hub TF (Fig. 2b). The TF encoded by *knotted1* (*kn1*), a master regulator of meristem maintenance including regulation of GA pathways[32], was also predicted as a conserved hub. In contrast, the uncharacterized TF DOF25 (*Zm00001d034163*) is potentially a transient hub connected to many genes in the 'leaf' GCN (degree centrality = 1) but fewer in the 'tassel' GCN (degree centrality = 0.57).

Since *lg2* mutants are strongly pleiotropic for TBN and LA phenotypes, we investigated tissue-specific connectivity of *lg2* in 'tassel' and 'leaf' GRNs. We observed substantial rewiring of its closest neighbor nodes with a small degree of edge preservation between "tassel" and "leaf", which included directed edges to *lg2* from TFs

EREB92 (*Zm00001d000339*) and ABI41 (*Zm00001d023446*) (Fig. 2c). In the "tassel" GRN, *lg2* was co-expressed with and co-regulated by several different TFs including TASSELSHEATH4 (TSH4), an interaction that was validated experimentally[33]. In the "leaf" GRN, predicted regulators of *lg2* were significantly enriched for HOMEOBOX (HB) ($q = 1.3e^{-04}$) and YABBY ($q = 2.38e^{-03}$) TFs, (Supplementary Data 14), and *lg2* was co-expressed not only with *lg1*, but also *liguleless related sequence1* (*lrs1*) and *sister of liguleless1*, closely related paralogs of *lg2* and *lg1*, respectively. Prior work investigating transcriptional networks at the blade-sheath boundary in maize similarly showed co-expression among these genes[34].

## Network motif analysis resolves context-specific topologies and core regulatory factors

To investigate the topology of the GRNs and interconnectedness of TFs, we annotated three-node network motifs; simple, recurrent regulatory circuits that appear in the network at a higher frequency than expected by chance[35]. We used information derived from directed edges inferred in the GRNs and systematically searched for three-node motifs (characterized by at least three edges) that were significantly overrepresented in the "tassel" and "leaf" GRNs. Out of ten possible three-node motifs, three types were significantly enriched in both networks ($Z$-score >20) and were present at a higher concentration (≥10%), namely the mutual-out, regulating-mutual and feed-forward loop (FFL) (Fig. 3a and Supplementary Data 15). To determine which TFs were predicted to be most influential in the transcriptional circuits, we ranked them based on frequency within annotated three-node motifs. We selected the top one hundred recurrent TFs from each of the "tassel" and "leaf" GRNs (Fig. 3b) for use in subsetting GWAS SNPs (described below). TFs belonging to ZHD ($q = 0.0028$) and HB ($q = 0.0002$) families were the most overrepresented in "tassel" and "leaf" GRNs, respectively.

FFLs have been described across organisms as local, repeated, and adapted genetic circuits[36]. To gain insight into the modularity of the FFLs within our context-specific GRNs, we identified the overrepresented TFs in the two parallel regulation paths controlled by nodes "X" and "Y" (Fig. 3c). GRAS, ZHD, and SBP were the most overrepresented TF families in the 'tassel' FFL$_{(X)}$ while HBs were overrepresented in the 'leaf' FFL$_{(X)}$. In both networks, these TFs were predicted to regulate several genes related to hormone biosynthesis, signaling, and nutrient sensing.

## A genomic selection approach to optimize phenotyping of maize diversity for architectural pleiotropy

We took a quantitative approach to identify loci that link TBN and LA in maize. To optimize selection for maximizing diversity in these architecture traits, a genomic best linear unbiased predictions (GBLUP) model was fit using the Goodman–Buckler diversity panel[37] and the predicted genomic estimated breeding values (GEBVs) for the

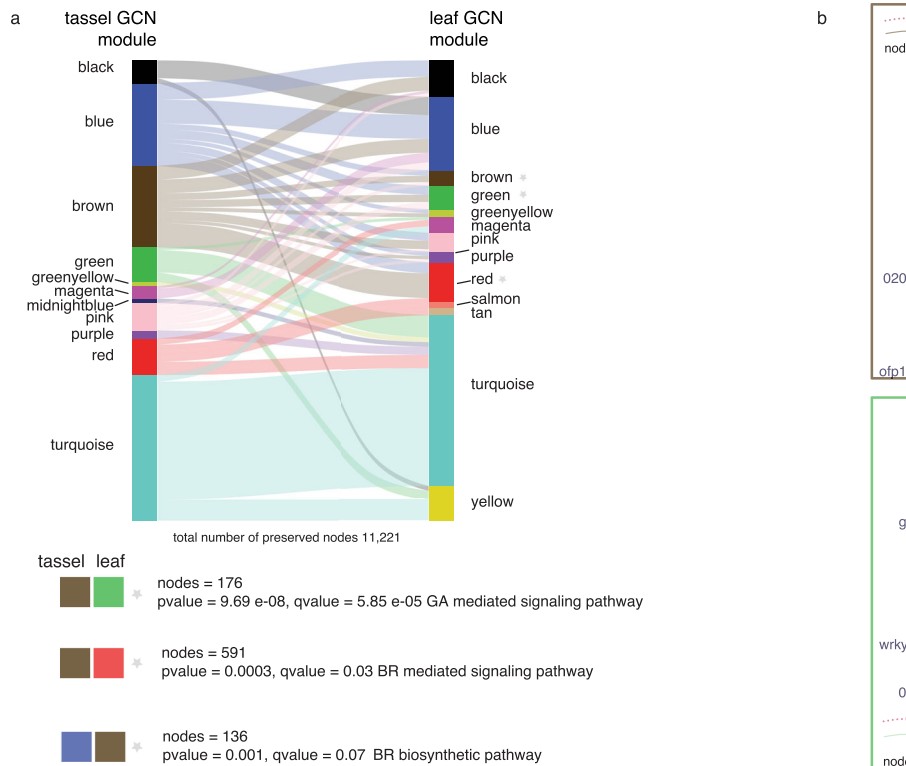

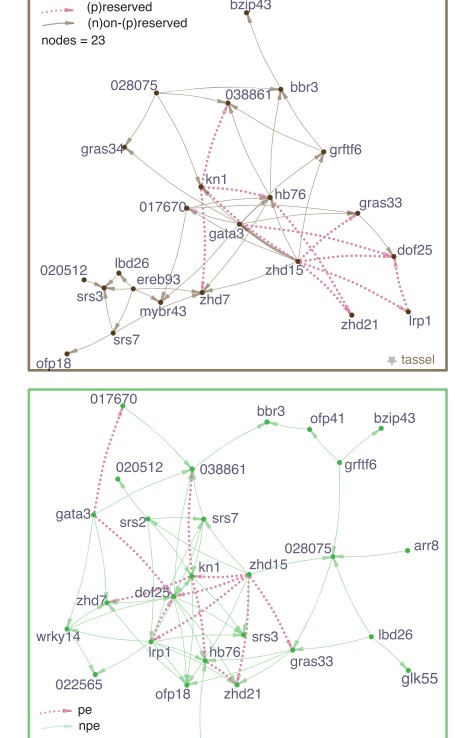

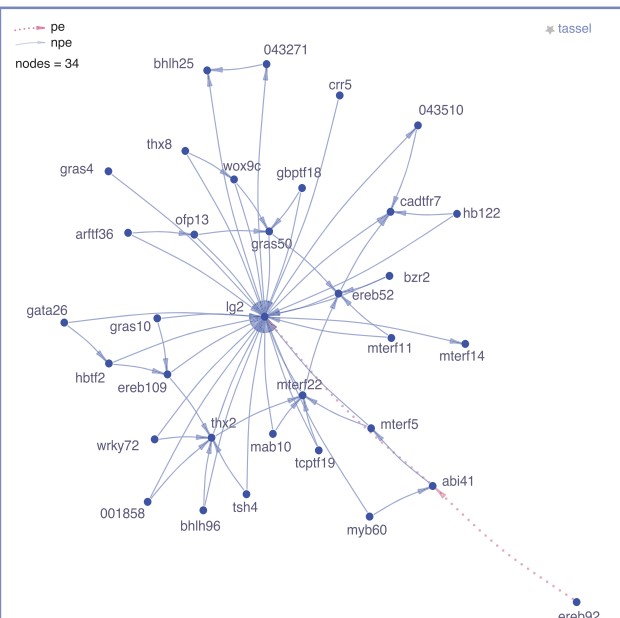

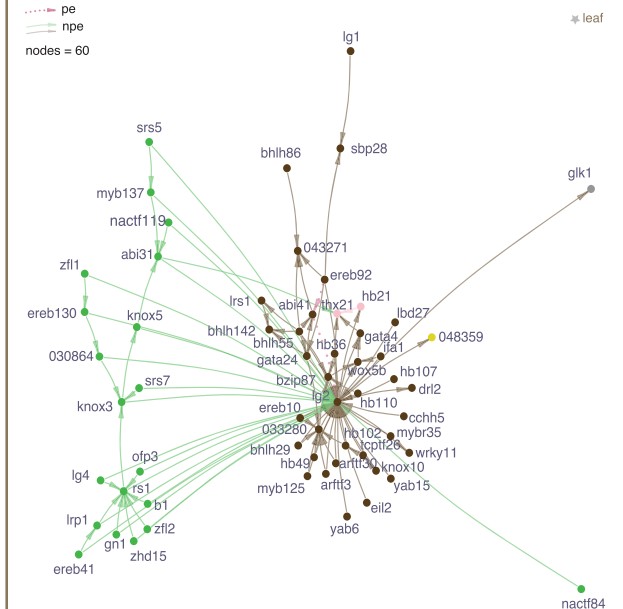

**Fig. 2 | Conserved and divergent network connections between tassel and leaf gene regulatory networks. a** Gene co-expression network (GCN) preservation between "tassel" and "leaf" networks defined from a permutation test ($n = 1000$ permutations). One-sided $p$ values are determined from the $Z$-score of individual measures under the assumption of normality. Preserved modules are indicated by color and connected by lines (width is proportional to number of preserved genes). Colored boxes beneath the plot represent examples of preserved gene co-expression sub-modules. **b** Topological graph representation of preserved TFs in the "tassel"-"leaf" "brown-green" sub-module based on data from GCN (edge weight) and gene regulatory network (GRN; edge directionality). Pink edges represent preserved regulatory connections, and brown or green edges represent the network-specific wires. **c** Topological graph representation of closest TF neighbors to *liguleless2* (*lg2*) in the 'tassel' and 'leaf' networks based on data from GCN (edge weight) and GRN (edge directionality). Pink edges represent preserved connections (pe preserved edges), and blue or brown/green edges represent network-specific wires (npe non-preserved edges). The edge length is proportional to the weight of gene co-expression. Nodes are colored based on their GCN module designation. Maize gene labels are from MaizeGDB or AGPv4 as Zm00001d$_{(6)}$. Source data are provided as a Source Data file.

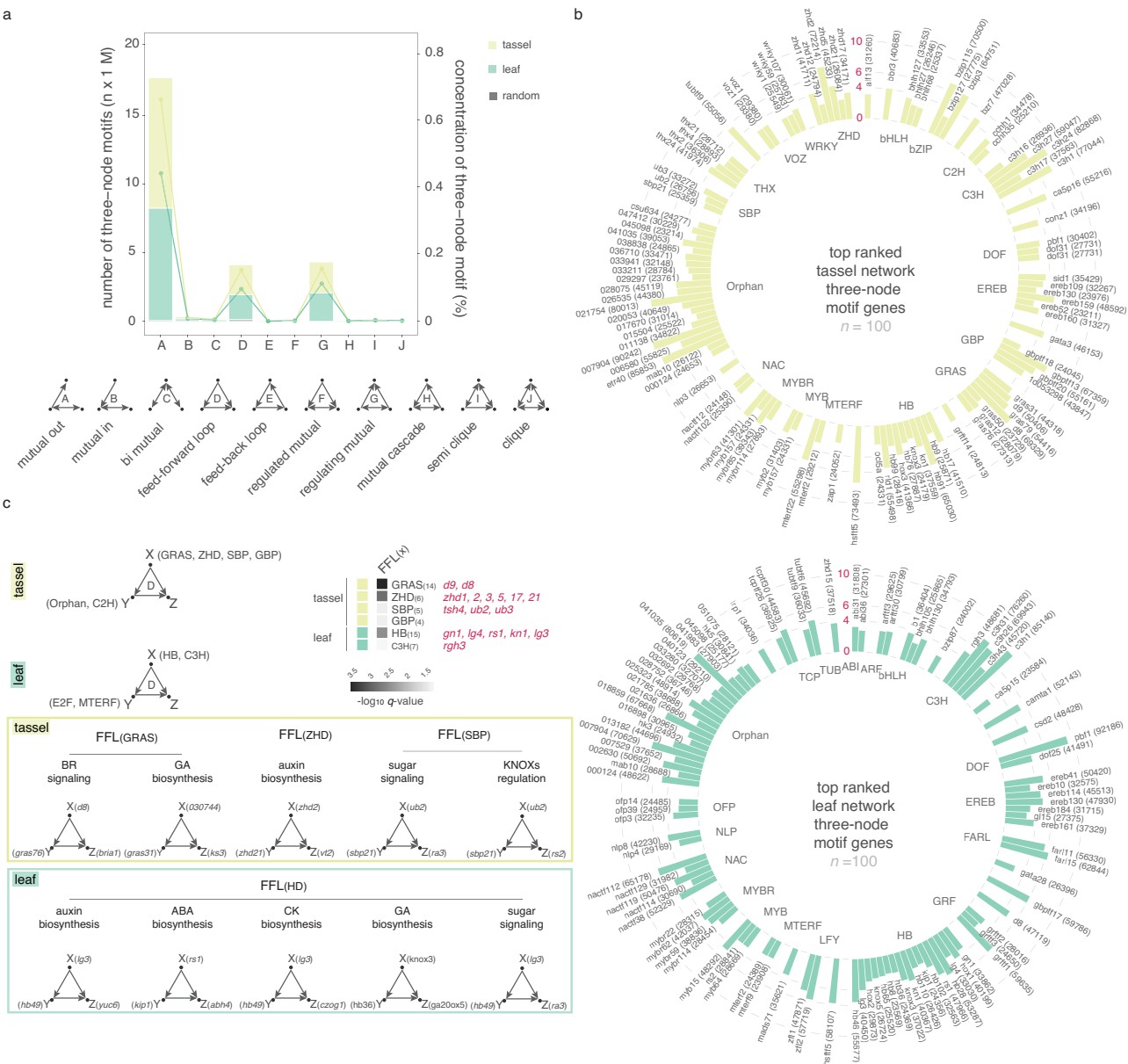

**Fig. 3 | Analysis of three-node network motifs. a** The number and type of three-node motifs identified in "tassel" and "leaf" GRNs compared to randomized networks ($n = 1000$ permutations). The x-axis represents the motif type (from A to J) of three-node motifs with edge number ≥3. **b** Top 100 ranked TFs based on their occurrence in the three-node motifs grouped by family. Bars represent TF frequency in the three-node motif (scale in red is TF frequency/10,000); non-normalized values are within parentheses. TF labels are from MaizeGDB or AGPv4 as *ZmO0001d*$_{(6)}$. **c** Examples of feed-forward loop (FFL) motifs mediated by select TFs. The heatmap displays overrepresented TFs in FFL$_{(X)}$, (examples of genes in each TF family are noted). Nodes represent genes, while the directed edges represent the potential regulatory relationships. Source data are provided as a Source Data file.

following traits in 2534 diverse lines from the Ames inbred panel[38]: TBN, LA, ear row number (ERN; included to maximize diversity for ear traits too), and first and second Principal Components of the phenotypes (PhPC1 and PhPC2, respectively) (Supplementary Data 16). Ames lines ($n = 1064$) were randomly selected to represent the distribution of the predicted PhPC1 and manually phenotyped for TBN and LA. Overall, the large portion of heritable variation among the selected genotypes resulted in prediction accuracies of 0.58 and 0.50 for TBN and LA, respectively. PhPC1 of the selected lines explained 56% of the total variance of TBN and LA, and was explained by trait loadings that were both positive. PhPC2 explained the remaining variance (44%) with positive (LA) and negative (TBN) loadings (Fig. 4a). These results collectively suggest that trait variation across maize subpopulations has diverged, including the pleiotropic components governing

TBN and LA. Interestingly, PhPC1 explained a larger portion of the variance than either TBN or LA, suggesting that PhPC1 may be a better source for detecting pleiotropy than testing phenotypes independently.

Because plant architecture has played a key role in maize adaptation, we hypothesized that genes underlying inflorescence and leaf architecture traits would be confounded with population structure. This hypothesis was underscored by how clearly the proxy traits PhPC1 and PhPC2 subdivided the lines into their respective subpopulations (Fig. 4a). Moreover, we observed a moderate correlation (0.53) between the PhPC1 (Supplementary Data 17) and published flowering time data[39]. Thus, the models we used in the ensuing analysis corrected for both population structure and familial relatedness (see Methods).

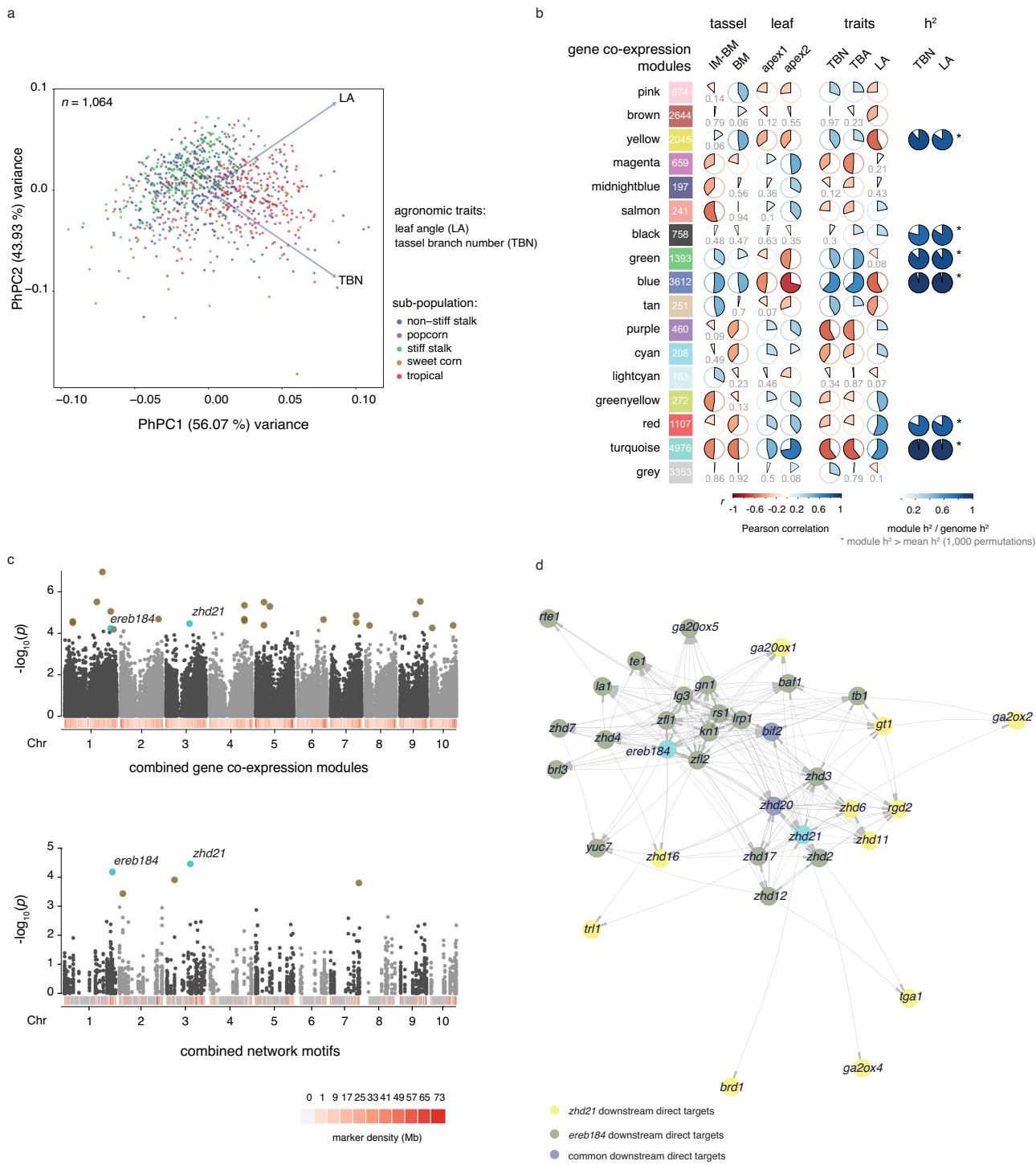

**Fig. 4 | Multi-trait GWAS detected candidate pleiotropic genes for TBN and LA.**
**a** Phenotypic principal components (PhPCs) of TBN and LA. Individuals are plotted according to their PhPC1 scores of TBN and LA on the x-axis and their PhPC2 scores on the y-axis. The percentage of total variance explained by a component is listed in parentheses on the axis. Blue arrows represent trait loadings for a given PhPC where its direction on the x-axis represents its contribution to PhPC1 and direction on the y-axis, its contribution to PhPC2. Sub-populations are color-coded and grouped according to PhPCs. **b** Analysis of module-trait relationships for the combined GCN. Modules are represented as colored boxes with the number of co-expressed genes indicated. The correlogram to the right represents the module-trait relationship where shades of blue and red represent positive and negative Pearson's correlation, respectively, with darker colors indicating a stronger positive or negative correlation. Modules representing more than 80% of the whole-genome heritability ($h^2$) for TBN and LA are indicated. **c** Multi-trait GWAS results using subsets of markers within 2 kb proximity of genes in the six co-expression modules with significant $h^2$ in panel b (top) and in the top 200 interconnected TFs within three-node motifs from Fig. 3b. Associations were evaluated using a two-sided likelihood ratio test from multivariate linear mixed models. Multiple testing correction was conducted on the number of SNPs in each partition using the Benjamini & Hochberg false discovery rate. Brown and blue dots represent significant SNP-trait associations; blue dots indicate loci identified in both analyses. **d** Topological graph representation of a sub-network including *ereb184*, *zhd21*, and their predicted downstream target genes based on data from the "leaf" GRN (edge directionality).

## Network-assisted multi-trait GWAS identifies genetic loci associated with TBN and LA

SNP markers were prioritized using gene sets related to TBN and LA based on network analyses (described below) and were used to guide multivariate GWAS. While this approach significantly reduced the marker density to varying degrees, we hypothesized that biological information derived from the networks could resolve novel SNP-trait associated markers.

To select genes for marker subsetting, we generated a comprehensive GCN using the entire RNA-seq dataset ($n = 140$) and identified 16 modules of co-expressed genes. Using a module-feature relationship analysis in WGCNA, we identified modules that had significant correlations with tissue types, developmental stages, or agronomic traits (Fig. 4b and Supplementary Data 18). For each GCN module, we estimated TBN and LA narrow-sense heritability ($h^2$) and compared the observed values to empirical distributions. Modules that fit the following three conditions were selected ($n = 6$): (i) significant module-trait correlation, (ii) observed $h^2$ values greater than the mean of their respective empirical distribution, and (iii) at least 80% of genome-wide $h^2$ explained (Fig. 4b and Supplementary Data 19). SNPs overlapping with genes in the selected six co-expression modules ($n = 106,790$; within 2 kb upstream and downstream of the gene model) accounted for nearly all of the genome-wide $h^2$ associated with TBN and LA (Supplementary Fig. 8). In addition to the combined GCN, we also used a reductionist approach by selecting the top one hundred recurrent TFs within the three-node network motifs derived from the "tassel" and 'leaf' GRNs (Fig. 3b) as an alternative way to partition SNP markers. Remarkably, SNPs ($n = 1972$) overlapping this group of genes still explained a significant portion of the genome-wide $h^2$, 57% and 70%, respectively for TBN and LA (Supplementary Fig. 8).

For each set of markers, from (1) combined GCN and (2) three-node motif analysis, we conducted single-trait GWAS for TBN, LA, and the first two PhPCs, and multi-trait GWAS analyses for TBN and LA. Our GWAS models collectively yielded 71 and 172 non-overlapping single-trait associated SNPs for the combined GCN and three-node motif sets, respectively (Supplementary Data 20). The majority (72%) of the trait associated markers in the three-node motif set were associated with PhPC2, which represents opposite trait loadings. Genes associated with PhPC2 were significantly enriched with GO terms implicated in meristem determinacy and maintenance (Supplementary Data 21).

Multivariate analysis with the two marker sets identified a total of 23 potential pleiotropic loci where SNPs were simultaneously associated with both TBN and LA phenotypes (Fig. 4c, Supplementary Fig. 9, and Supplementary Data 22). Two of these SNPs were common in analyses with both marker sets indicating high-confidence associations that may contribute to phenotypic pleiotropy. Both SNPs were located within genes encoding TFs; one in the last exon of *ereb184* (*Zm00001d034204*), an ortholog of *AINTEGUMENTA1* in *Arabidopsis thaliana* known to control plant growth and floral organogenesis[40], and one in the last exon of *zhd21* (*Zm00001d041780*), which was shown to express in leaf primordia during ligule initiation[34]. These two markers were also associated with TBN in the single-trait GWAS. Also, the marker in *ereb184* associated with PhPC1 in the single-trait GWAS, reinforcing the notion that PhPC1 may be used to detect pleiotropic loci. To further validate the associations at *ereb184* and *zhd21*, we performed candidate gene association analysis using the maize Nested Association Mapping (NAM) recombinant inbred line (RIL) families along with their publicly available phenotype data for LA and TBN[19,41]. This analysis showed peak SNP-trait associations in several NAM families (Supplementary Fig. 10).

We found that *ereb184* and *zhd21* were connected in the "leaf" GRN through four TFs: *ereb93, zhd20, zhd16*, and *barren inflorescence2*; the latter regulates tassel branch outgrowth in maize[42]. Predicted downstream targets of these two TFs suggest that they regulate architecture traits through different but interconnected

developmental circuits made of several other *zhd* genes, *knox* genes, and genes regulating hormone metabolism, transport, and signaling, especially in auxin, GA and BR pathways (Fig. 4d and Supplementary Data 23). The link between *ereb184* and *zhd21* was also supported by statistical epistasis analysis (Supplementary Fig. 11 and Supplementary Data 24), which highlighted their interaction together with other TFs and members of the ZHD family, such as *zhd1, zhd12, and zhd15*.

## Maize network data guide explorations of SNP-trait associations in sorghum

We further tested whether the small set of markers selected based on three-node network motifs in maize was sufficient to guide GWAS to candidate SNP-trait associations for LA in sorghum. Among the maize genes within the motif set, we identified 146 sorghum orthologs and their associated SNPs from the Sorghum association panel (SAP)[43]. Using the selected SNP markers together with publicly available LA phenotype data for the SAP[44], we first tested $h^2$ for this trait and found that variation in these genes explained more than 60% of the sorghum genome-wide LA $h^2$ (Fig. 5a). Therefore, we conducted a single-trait GWAS for LA. Eight sorghum LA-associated markers were found within or proximal to seven genes (Fig. 5b and Supplementary Data 25), including orthologs of maize genes *nac112* (*Sobic.009G143700*), *ofp39* (*Sobic.008G042200*), *c3h16* (*Sobic.006G256500*), and *Zm00001d026535* (*Sobic.006G254500*), which were also identified in maize single-trait GWAS for PhPC1, PhPC2 and/or TBN. Additionally, a significant SNP was identified within the maize ortholog of *ereb114* (Sobic.002G022600), a paralog of *ereb184*. Based on first neighbors' connectivity in the maize "leaf" GRN, *ereb114* is situated either up- or downstream of known *knox* genes involved in meristem maintenance and axillary meristem formation in maize, e.g., *kn1, gn1, rs1*, and *lg4*, suggesting conservation of developmental circuits controlling LA between maize and sorghum (Fig. 5c). Furthermore, *ereb114* is predicted to directly target *zhd21*, another high-confidence gene candidate from the multi-trait GWAS for LA and TBN.

## *zhd* genes modulate tassel and leaf architecture

Throughout this study, genes encoding ZHD TFs repeatedly emerged as important players in tassel branch and ligule development. These TFs appear as putative hubs in the networks or are highly connected to known developmental pathways. *zhd* genes are enriched within BR signaling-responsive genes during early development, and *zhd21* was identified as a high-confidence candidate gene in multi-trait GWAS for TBN and LA. The extensive interconnectedness among numerous ZHD family members suggests functional redundancy and potential for molecular fine-tuning. To test whether disruption of network candidate *zhd* genes have a phenotypic effect on tassel and/or leaf architecture, we studied the effects of independent *Mutator* (*Mu*) transposon insertions in the coding regions of *zhd21* and *zhd1* (UniformMu, Methods). One allele of *zhd21* (*zhd21-1*; mu1056071) showed significantly fewer tassel branches and a more upright LA compared to W22 normal plants (Wilcoxon *p*-value TBN = 2.1e$^{-04}$, LA = 8.5e$^{-09}$), while a different allele of *zhd21* (*zhd21-2*; mu1018735) and *zhd1* (mu1022277) showed no significant difference in either trait (Fig. 6a). The mutation in *zhd21-1* disrupts the zinc-finger domain, which we expect confers the observable phenotype (Supplementary Fig. 12). Since we hypothesize some degree of functional redundancy among *zhd* genes, lack of phenotypes for the other alleles was not surprising.

We tested whether stacking *zhd21* and *zhd1* alleles with no apparent phenotypes in homozygous single mutants resulted in architectural differences. Interestingly, plants that were homozygous for the *zhd1* mutant allele and heterozygous for *zhd21-2* had significantly more tassel branches and more upright LA compared to plants heterozygous for both alleles (Wilcoxon *p*-value TBN = 0.0094, LA = 0.0008) (Fig. 6b, c). Plants homozygous for *zhd21-2* and heterozygous for *zhd1* showed no significant difference in either trait. Our results suggest that *zhd21* and *zhd1* influence tassel branching and leaf

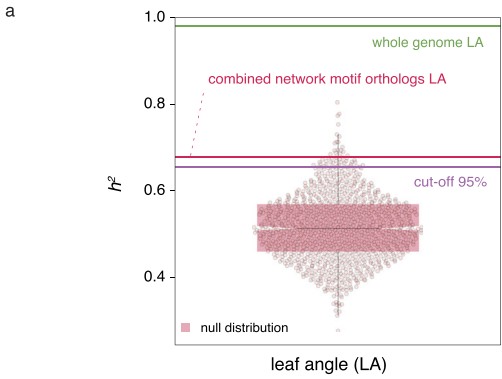

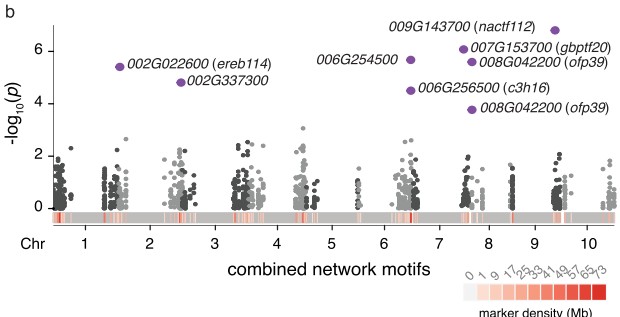

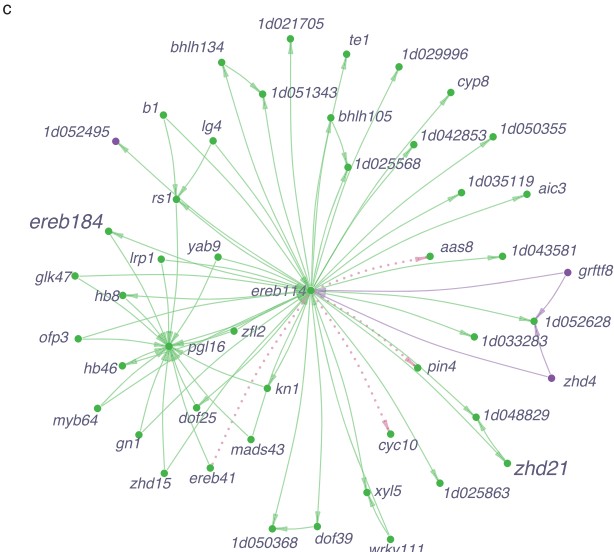

**Fig. 5 | GWAS for LA in sorghum using biological information derived from maize. a** LA narrow-sense heritability ($h^2$) of 146 sorghum-maize syntenic orthologs in comparison to a null distribution of $h^2$ ($n = 1000$ permutations). The violin plot represents the null distribution, including the median (black bar), first and third quartiles (shaded pink), and the minimum and maximum values. The purple line represents the 95th percentile of the null distribution, the red line represents LA $h^2$ of the sorghum orthologs, and the green line is the whole-genome LA $h^2$. **b** Manhattan plot showing GWAS results for LA based on markers in the proximity of the 146 sorghum-maize syntenic orthologs. Purple dots represent significant SNP-trait associations for LA with sorghum gene IDs ($Sobic._{(10)}$) noted. Associations were evaluated using a two-sided $F$-test. **c** Topological graph representation of *ereb114* and its closest network neighbors in the maize 'leaf' GCN (edge weight) and GRN (edge directionality). Green and purple nodes and edges are colored based on module assignment in the 'leaf' GCN. Pink edges represent preserved connections with the "tassel" GCN. Edge length is proportional to the gene co-expression weight. Source data are provided as a Source Data file.

architecture, and that complex network connectivity among *zhd* family members may allow for precise modulation of pleiotropy in these traits through combinations of alleles.

### Structural variation in the promoter of *ereb184* modulates pleiotropy in TBN and LA

Our gene candidate *ereb184* was among the top hundred recurring TFs within 'leaf' GRN three-node motifs and positioned as a regulatory factor in many FFLs ($FFL_{(x)}$; $n = 7422$). Notably, these FFLs were over-represented by ZHDs at the $FFL_{(y)}$ node (Fig. 7a). We found that *ereb184*-regulated FFLs potentially regulate a set of downstream genes ($FFL_{(z)}$) associated with functional categories that overlapped with those enriched during tassel branch and ligule development (Fig. 7a and Supplementary Data 26). For example, four KN1-regulated FFLs (validated by ChIP-seq data[32]) included *ereb184* connections to several known homeotic genes that were predicted in the 'leaf' GRN to directly target *lg2* (Fig. 7b).

Scanning the gene regulatory space around *ereb184*, we observed ~100 kb of intergenic sequence upstream of the TSS. We integrated publicly available omics' datasets to investigate potential regulatory regions. We identified three genomic regions with regulatory signatures, including unmethylated marks, which colocalized with conserved non-coding sequences and showed increased nucleotide diversity, similar to that overlapping the coding region of *ereb184* (Fig. 7c). To investigate tissue-specific chromatin accessibility, we generated Assay for Transposase-Accessible Chromatin (ATAC)-seq data from the B73 shoot apex 2 tissue (including developed ligules) and compared the profiles in this region to publicly available ATAC-seq data from tassel primordia[45]. All three regulatory regions showed tissue-specific chromatin signatures that overlapped with other epigenetic marks, highlighting potential tissue-specific regulation of *ereb184*. Overlaying publicly available maize leaf chromatin interaction data (Hi-C)[46] predicted chromatin looping between the *ereb184* promoter and one of the upstream intergenic regions (Fig. 7c).

Using public data from the maize NAM Consortium[47], we identified structural variation (SV) of approximately 5 kb in the promoter region of *ereb184*. Based on resequencing data of the 26 NAM founder lines[47], this SV was either present or absent, and mainly absent in tropical inbred lines. We also observed tissue-specific chromatin accessibility adjacent to the SV, suggesting it is located within a regulatory region. The marker identified through our network-guided GWAS in the genic region of *ereb184* was in linkage disequilibrium (LD) with the SV (Supplementary Fig. 13). This result was supported by a multivariate candidate gene association analysis at the *ereb184* locus for TBN and LA using whole-genome SNPs[48], which also identified a peak-associated SNP in the SV ($-\log_{10}$ p-value = 7.24).

We tested whether the presence/absence variation (PAV) of the SV in the promoter of *ereb184* influenced pleiotropy between TBN and LA. In addition to NAM founder reference genomes, we extended the analysis to include 216 other accessions that we phenotyped from the Goodman–Buckler panel[37], incorporating existing sequencing data[49] to define PAV (Supplementary Fig. 14). Strikingly, we found that maize lines without the SV showed increased TBN and had more upright leaves, while lines with the SV showed the opposite trend, and these results were significant by Wilcoxon rank-sum test (p-values TBN = 0.008, LA = 0.044) at $\alpha = 0.05$. Also, based on transcriptome data from the NAM Consortium, founder lines with the SV present in the *ereb184* promoter had lower gene expression (Wilcoxon rank-sum test $p = 0.029$) in shoot tissue (Fig. 7c). These results show that SV in the *ereb184* promoter contributes to the regulation of these pleiotropic, agronomic traits.

## Discussion

In maize, mutants that affect the ligule also affect tassel branch initiation, and genes expressed at initiating ligules are also expressed

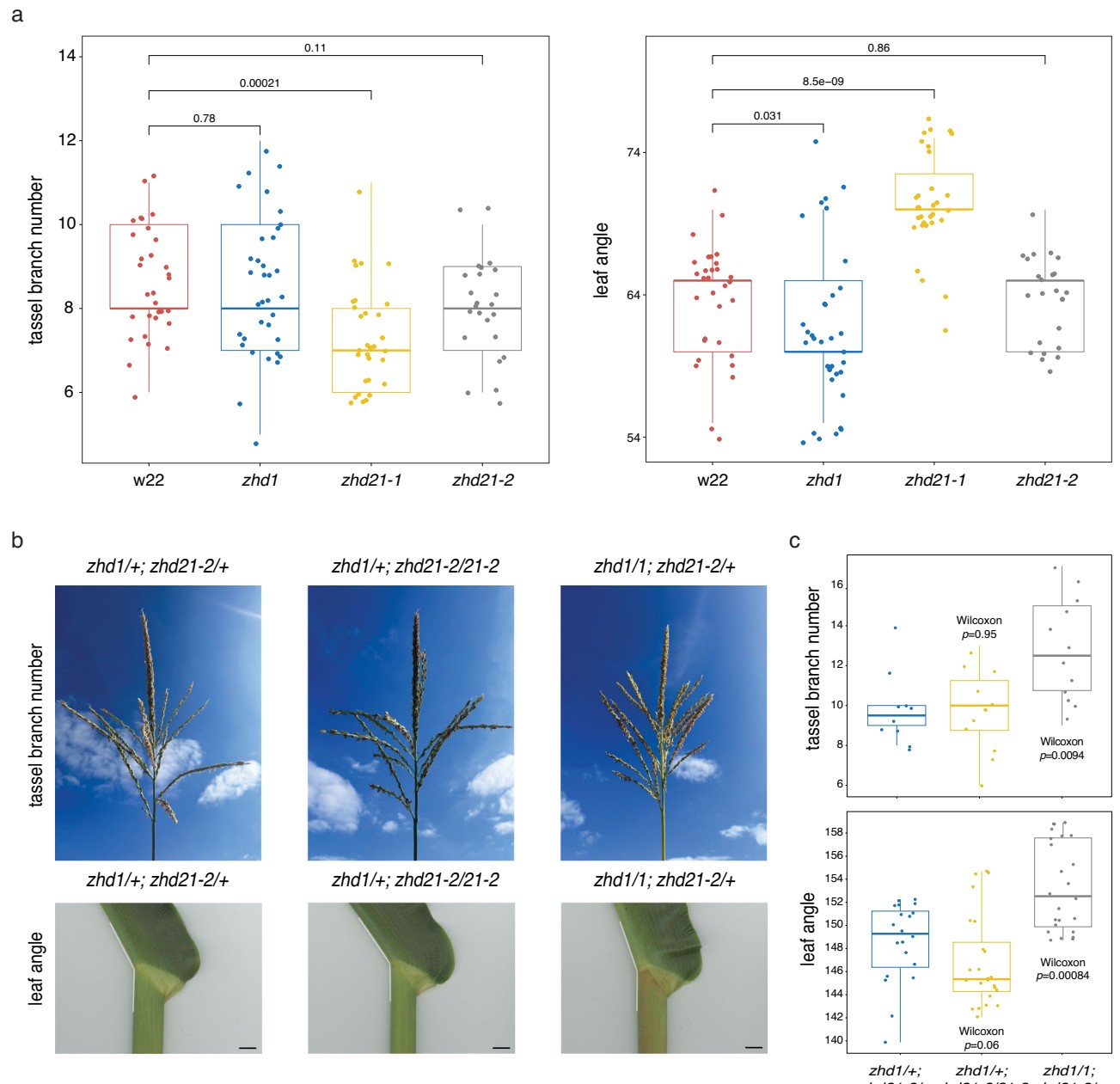

**Fig. 6 | *zhd* mutant alleles contribute to tassel and leaf architecture.**
**a** Phenotypic characterization of TBN and LA in homozygous *zhd* alleles compared with w22 control plants. **b** Representative tassel branch and leaf angle phenotypes resulting from genetic interactions between mutant alleles of *zhd1* and *zhd21*, from left to right: *zhd1/+;zhd21-2/+*, *zhd1/+;zhd21-2/zhd21-2*, *zhd1/1;zhd21-2/+*. Black bar = 1 cm. **c** Among combinations of mutant alleles,

*zhd1/1;zhd21-2/+* showed significant differences in TBN and LA compared to *zhd1/+;zhd21-2/+* normal siblings. Box plots in (**a**, **c**) represent phenotypic values in a given genetic background and indicate median (thick bar), first and third quartiles, and minimum/maximum values. *P* values were calculated based on a one-sided Wilcoxon rank-sum test. Source data are provided as a Source Data file.

at the boundary of other lateral organs[10,11] such as tassel branches. In this study, we leveraged this well-characterized maize genetics system to investigate the molecular underpinnings of pleiotropy: network rewiring around pleiotropic factors in two developmental contexts, the redeployment of TFs for analogous but different developmental processes, and *cis*-regulatory variation modulating pleiotropic loci. By strategically integrating context-specific biological data and multivariate GWAS models that exploit maximal diversity in TBN and LA, we identified new regulatory factors contributing to architectural pleiotropy in maize. Our approach can be applied in any genetic system to disassociate pleiotropic phenotypes through the manipulation of *cis*-regulatory components and gene network connections.

Pleiotropy in crop phenotypes can limit productivity ceilings. For example, the selection of a certain desired phenotype may come with unintended deleterious manifestations of another. This can happen when a gene controlling a target trait also functions in another developmental or physiological context. In recent years, the dramatic production gains of the 20th century have plateaued in the world's most important cereal crops[50]. Looking forward, step changes in crop improvement and sustainability will rely on targeted manipulation of regulatory pathways to fine-tune agronomic traits for enhanced plant productivity and resilience in dynamic environments[51,52]. Central to this is the ability to predict and design highly specific genetic changes at pleiotropic loci with minimal perturbation to the complex networks

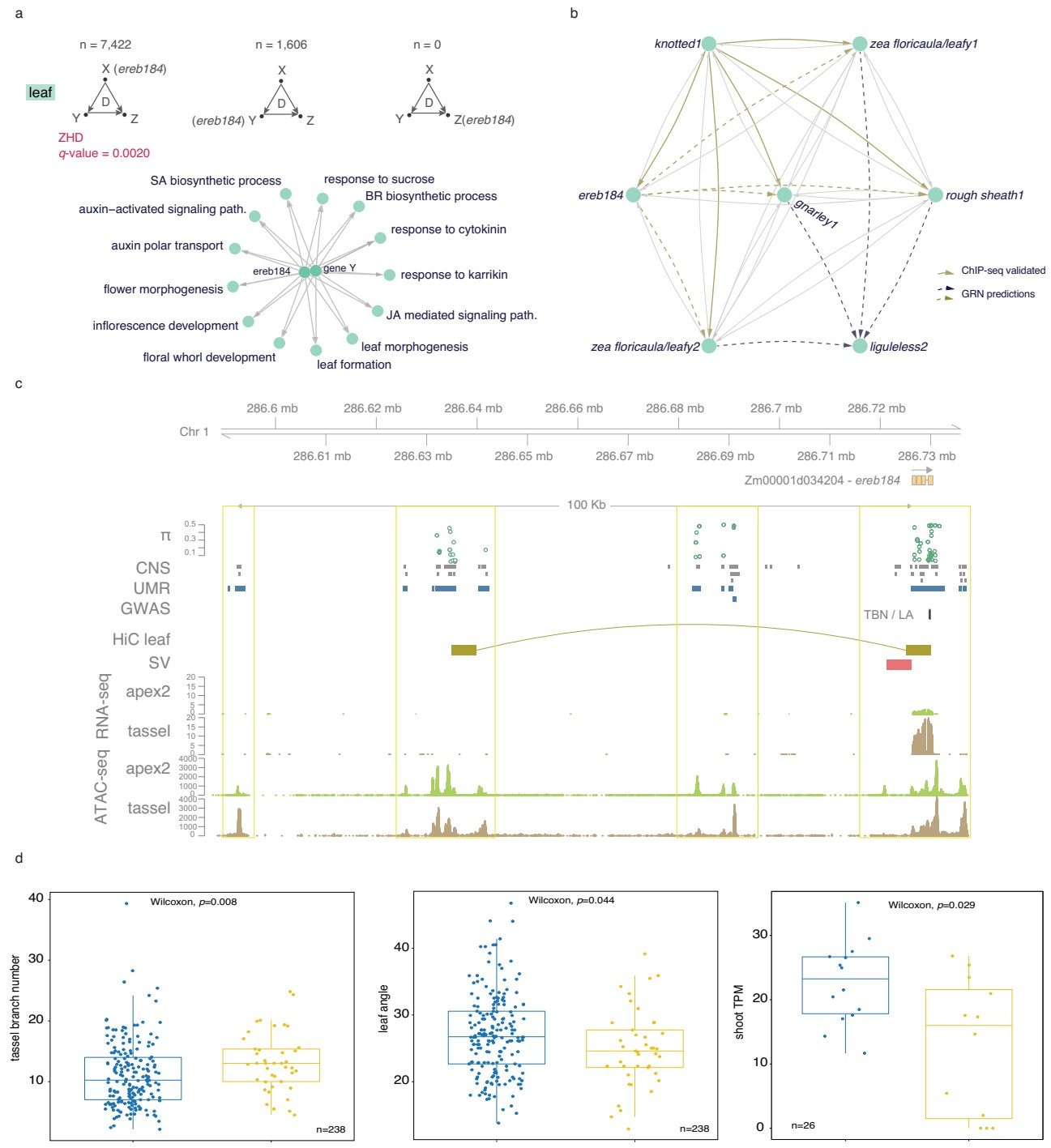

**Fig. 7 | *ereb184* modulates pleiotropy in TBN and LA. a** FFLs from the 'leaf' GRN are represented with *ereb184* in each of the node positions and the number of times each occurs. The graph below shows GO terms overrepresented within the Z position when EREB184 is in the X position. **b** An example of a sub-network from the 'leaf' GRN shows *ereb184* participating in multiple FFLs with KN1 and other proximal-distal patterning TFs that target *lg2*. The solid yellow arrows represent ChIP-seq validated connections. **c** Genomic view of the intergenic space upstream (100 kb) and just downstream of *ereb184*. The tracks from top to bottom: nucleotide diversity (π), conserved non-coding sequences, multi-trait GWAS, unmethylated regions, leaf chromatin interaction data (Hi-C), structural variation, RNA-seq, and ATAC-seq. Data from public sources are indicated in Data accessibility. **d** The effect of structural variation (SV) presence/absence in the promoter region of *ereb184* across 238 accessions on TBN and LA traits, and the effect of SV on *ereb184* expression within 26 NAM founder lines. Box plots represent the median, first and third quartiles, and minimum/maximum values. *P* values were calculated based on a one-sided Wilcoxon rank-sum test. Source data are provided as a Source Data file.

in which they reside. Therefore, knowledge of context-specific gene networks and functional *cis*-elements will enable greater precision in engineering or breeding optimal plant ideotypes.

In comparing predicted gene regulatory interactions between tassel branch and ligule development, both conservation and rewiring of the GRNs were observed. For example, KN1 and ZHD15 were maintained as hubs with conserved regulatory connections in both developmental contexts. Conservation around KN1 was not unexpected given its role as a master regulator of meristem maintenance. Identifying ZHD15 as a conserved hub was novel, but consistent with several

lines of evidence in this study that implicate ZHDs as central players in architectural pleiotropy between tassel branching and leaf architecture. Alternatively, there was extensive rewiring of predicted network connections around LG2, which strongly controls TBN and LA in a pleiotropic manner. This suggests that the redeployment of LG2 in different developmental contexts was likely accompanied by the loss and gain of regulatory connections to gene targets and targeting TFs. Perhaps LG2 serves a common function in setting up boundaries, and its interactions with context-specific sub-networks shape the specific developmental programs.

We showed that these biological context-specific networks could be used to subset markers for multi-trait GWAS and identify new loci that contribute to architectural pleiotropy in TBN and LA. Using two different network-guided approaches we identified several gene candidates in and around significant SNP-trait associations. Strikingly, both approaches identified significant SNPs within two of the same TF-encoding genes, *zhd21* and *ereb184*. Both TFs are members of large families that are largely uncharacterized in maize, and our results showed that ZHD and EREB TFs were highly interconnected within the regulatory networks controlling tassel and leaf architecture. Notably, while *zhd21* was down-regulated in *lg2* tissue samples enriched for developing ligules, there was no significant difference in its expression in *lg2* tassels, which make few to no tassel branches, potentially consistent with *zhds* regulating pleiotropic effects between these traits. Furthermore, stacking *zhd* alleles with no observable phenotypes in these traits resulted in pleiotropic phenotypic expressions. This suggests functional redundancy among *zhd* genes and that further analyses of genetic interactions among *zhd* mutant alleles should provide insight into how to precisely manipulate plant architecture.

The vast stretch of intergenic space (~100 kb) upstream of *ereb184* is sprinkled with regulatory signatures, including tissue-specific accessible chromatin regions, which potentially regulate its context-specific expression. The SV in its upstream promoter region, which is present in approximately half of the NAM founders, appears to control the expression of *ereb184*; expression levels are significantly different in NAM lines with the SV compared to those that don't. The presence or absence of the SV may contribute to differences in spatiotemporal expression patterns of *ereb184* and underlie the shifts in pleiotropy between TBN and LA. The origin of this SV is unclear. A blast search of the SV DNA sequence against a curated transposable element database revealed the presence of a probable LTR Gypsy retrotransposon (86% alignment identity) in the 3-prime region of the SV, as well as interspersed Helitron fragments. Differential chromatin accessibility between tassel primordia and shoot apex was evident in a regulatory region immediately proximal to the SV. Perhaps the SV is disrupting TF-binding interactions in the *ereb184* promoter or even interfering with long-range regulation through chromatin looping, as suggested by a proximal Hi-C interaction.

Small effect loci for target traits are difficult to pinpoint with GWAS, given that large trait effects generally dominate, which is compounded by the threshold for significance rising with marker density due to multiple corrections. Targeted GWAS with a limited, but biologically informed marker set, has been demonstrated using pathway enrichment analysis[53] and chromatin accessible regions[54]. Application of these methods to identify informative smaller SNP sets can enhance genomic predictions. Here, we showed that even SNP subsets identified with small numbers of genes prioritized through network analyses not only produced heritabilities similar to a genome-wide set of markers, but also enabled the detection of small effect loci that were validated to contribute to both tassel and leaf architecture. Notably, we further showed that the context-specific networks from maize could be used to inform GWAS for LA in closely related sorghum. The analysis in sorghum identified *ereb114*, a paralog of *ereb184*, which was connected to it and *zhd21* in the network.

Our analyses support the utility of biologically informed, context-specific networks for guiding GWAS in various applications and for identifying loci that connect phenotypic traits. Results highlight various mechanisms by which the expression of pleiotropic trait phenotypes are modulated, including through network interconnectedness of functionally redundant TF family members or SV in *cis*-regulatory components. We anticipate that this approach can be used widely to identify pleiotropic loci for manipulation in crop improvement.

## Methods

### Plant material for RNA-seq experiments

Mutants were introgressed at least five times into the maize B73 inbred. Mutant alleles used were: *brassinosteroid insensitive2* (*bin2*-RNAi), *brassinosteroid insensitive1* (*bri1*-RNAi), *feminized upright narrow* (*fun1-1*), *liguleless1* (*lg1-R*), *liguleless2* (*lg2-R*), *ramosa1* (*ra1-R*), *ramosa2* (*ra2-R*), *wavy auricle blade1* (*wab-rev*, *Wab-1* – dominant). All were grown along with B73 controls in environmentally controlled growth chambers at the Danforth Center Integrated Plant Growth Facility with 14 h days, 28/24 °C, 50% humidity, and 450 μmol light. Plants were sown in cone trays (5 cm diameter, 11.5 cm depth, 142 mL total volume) in a Metromix 360-turface blend. At 14 days after sowing (DAS), seedlings were transplanted to larger pots (27 cm diameter, 24 cm depth, 14 L total volume; three plants per pot) with Berger 35% soil and 10 g of corn top dressing. In a trial experiment prior to tissue sampling, the development of tassel primordia was tested and staged for uniformity across genotypes. Plants were staggered over two weeks for tissue collections in identical conditions.

### Tissue sampling and RNA extraction

Tassel primordia were hand-dissected and flash-frozen in liquid nitrogen. For each genotype, fifteen primordia were pooled per replicate for stage 1 and ten for stage 2 (Supplementary Fig. 1). To collect shoot apex 2, whorls were removed to the node from 21 DAS plants until the ligule of a developing leaf was ~0.75 cm from its node. Three to four whorls were then excised, and a 2 mm section of leaf surrounding the ligule region of each of these leaves were collected. From the remaining tissue, shoot apex 1 was sampled by cutting an additional 2 mm section to include the base of remaining developing leaves and the region, including the shoot apical meristem. Two to three individuals were collected per replicate and the material was flash-frozen immediately after dissection.

For each tissue type and developmental stage, four biological replicates were collected. Tissue samples were ground using a bead shaker with liquid nitrogen in a 2 mL tube with a 5 mm ceramic bead. RNA isolation from tassel material was performed using the PicoPure RNA isolation kit (Thermo Fisher Scientific) according to the manufacturer's instructions with the following adjustments: 40 and 60 μL of RNA extraction buffer were added, respectively, to stage 1 and stage 2 ground tassels. After 30 min incubation at 42 °C, samples were centrifuged at 800×*g* for 2 min. An equal volume of 70% EtOH was added to samples and processed according to kit directions. On-column DNaseI treatment was performed per instructions using an RNase-Free DNaseI kit (Qiagen) to remove residual DNA. RNA isolation from shoot apex samples was performed using the Zymo quick-RNA plant kit according to manufacturer instructions with the following adjustments: Lysis buffer was added directly to the ground tissue and centrifuged, and supernatant was added directly to the filtration column. DNaseI treatment was performed using the supplied DNaseI enzyme according to manufacturer instructions. RNA was quantified using the NanoDrop One Spectrophotometer (Thermo Fisher Scientific) and the RNA-6000 Pico chip from (Agilent) to ensure RNA integrity.

### RNA-seq libraries, sequencing, and data analysis

Poly(A)⁺ RNA-seq library preparation and sequencing were outsourced to Novogene (USA). Libraries were multiplexed 12/lane and sequenced

using the Illumina HiSeq4000 platform with a 150-bp paired-end design. On average more than 60 million paired-end reads were achieved per sample with quality score (Phred-score) >30. Raw reads were processed to filter out low-quality reads, adapters or barcode remnants using Cutadapt v2.3[55] and the wrapper tool TrimGalore v0.6.2 with default parameters except for –length 70 –trim-n –illumina. Clean reads were used to quantify the maize B73 AGPv4 gene models with Salmon v1.4.0 using the selective alignment method with a decoy-aware transcriptome[56]. A Salmon index was created using default parameters from the cdna fasta file (Zea_mays.AGPv4.cdna.all.fa) together with the genome reference fasta file (Zea_mays.AGPv4.dna.toplevel.fa) to generate the decoys for the selective alignment method. Maize reference files were downloaded from Ensembl Plants release 34.

Briefly, clean reads were mapped to the reference transcriptome using the Salmon quant command with default parameters except for options -l A –numBootstraps 100 –validateMappings. Expression levels were imported in R using the Bioconductor package tximport[57], summarized to gene level using the function summarizeToGene(), and presented in TPM (transcript per kilobase million). Overall gene expression levels between replicated samples ($n = 4$) were highly related with correlation coefficients $r \geq 0.92$.

Differential expression analysis was performed using the Bioconductor package DESeq2[58]. Pairwise contrasts were applied to compare mutant genotypes against equivalent normal samples. To test differences along the tassel developmental gradient attributable to a given genotype in comparison with B73, we set up an interaction design formula: ~ Genotype + Tissue + Genotype:Tissue. Genes were considered differentially expressed based on a false discovery rate ≤0.05.

To standardize the relative expression of each gene across the *bin2*, *bri1*, and B73 control genotypes, we normalized the expression values for each gene within the triad as follows:

$$\text{Relative expression } bin2_{\text{gene(i)}} = \text{TPM}(bin2_{\text{gene(i)}})/[\text{TPM}(bin2_{\text{gene(i)}}) + \text{TPM}(bri1_{\text{gene(i)}}) + \text{TPM}(B73_{\text{gene(i)}})] \quad (1)$$

$$\text{Relative expression } bri12_{\text{gene(i)}} = \text{TPM}(bri1_{\text{gene(i)}})/[\text{TPM}(bin2_{\text{gene(i)}}) + \text{TPM}(bri1_{\text{gene(i)}}) + \text{TPM}(B73_{\text{gene(i)}})] \quad (2)$$

$$\text{Relative expression } B73_{\text{gene(i)}} = \text{TPM}(B73_{\text{gene(i)}})/[\text{TPM}(bin2_{\text{gene(i)}}) + \text{TPM}(bri1_{\text{gene(i)}}) + \text{TPM}(B73_{\text{gene(i)}})] \quad (3)$$

Expression time (ET) was calculated using a smooth spline regression model[59] with the R function bs(). We fitted a b-spline (3-knot with three degrees of freedom) modeled on the first and second PC of the 500 most dynamically expressed genes across normal tassel development. Data points from mutant backgrounds were classified based on their location on the spline in relation to this model.

## Gene network analyses

GCNs were built using the R package WGCNA (v.1.68)[60]. Expression data of protein-coding genes were imported into R with the function DESeqDataSetFromTximport(). For each GCN, we selected expressed genes based on row mean >5 counts with the R function rowMeans() and normalized the count expression level of each gene according to the variance stabilizing transformation (VST) with the function vst() from DESeq2 package. Pearson correlation was used to select samples for the gene co-expression networks. Highly correlated biological replicates with $r \geq 0.92$ were retained and independently input in the network analyses. Based on the correlation coefficient, only one sample derived from the *lg1-R* stage 1 tassel (replicate 2) was excluded from the network analyses.

The soft power threshold was set to 6 for the "tassel" and "leaf" GCNs and to 7 for the combined network. Module detection was calculated via dynamic tree cutting using the function blockwiseModules() with the following parameters: type = signed, corType = bicor, minimum module size = 30, mergeCutHeight = 0.25. The parameter maxBlockSize for each network was set equal to the total number of expressed genes passing the mean cutoff as described above; 22,499 and 22,716, respectively, for "tassel" and "leaf" GCNs. The topographical overlap matrix (TOM) was calculated for each network using the function TOMsimilarityFromExpr() with parameters matching those used in the module detection. Networks were exported using the function exportNetworkToCytoscape() with parameters: weight = TRUE and threshold = 0.00. The R package igraph v1.2.4.1[61] was used to build graphs from exported networks with the function graph_from_data_frame() and to calculate the graph statistics. Preserved modules between "tassel" and "leaf" GCNs were computed using the function modulePreservation() with 1000 permutations. Gene submodule preservation between networks was calculated using the R package GeneOverlap (v.1.28).

For the combined GCN generated from all samples, the module-to-sample association analysis was conducted to evaluate the correlation between the module eigengene and samples of different developmental groups: (i) tassel primordia at stage 1, (ii) tassel primordia at stage 2, (iii) shoot apex 1, and (iv) shoot apex 2. In addition, we tested associations between module eigengene and three traits: LA, TBN, and TBA. A metafile was created where samples were categorized according to the four sample groups and three traits. The R function cor() and corPvalueStudent() were used to test the correlation between the module eigengene and the variables. Modules with $r > |0.8|$ were considered strongly correlated.

For the context-specific GRNs, a machine learning approach was applied to predict targets of known maize TFs using the Bioconductor package GENIE3[62]. Maize TFs were downloaded from the GRASSIUS[63] repository and overlapped with the expression matrices. TFs were set as "regulators" to infer their "targets" based on gene expression abundance. GENIE3 was run with parameters: treeMethod = "RF", nTrees = 1000 and putative target genes were selected with a weight cutoff ≥0.005. DAP-seq data[64] for ZHD TFs supported our predictions with accuracies ranging from 47 to 70%, exceeding the 95th percentile threshold of a null distribution derived from 1000 random permutations.

To determine the enrichment of three-node subgraphs in the GRNs, we scanned for all possible three-node subgraphs and compared results to a set of randomized networks ($n = 1000$) with the same number of nodes and edges. This analysis was conducted with R package igraph v1.2.4.1 using the following functions: graph.full() with $n$ equal to the number of genes in the context-specific GRNs, triad.census(), cliques() with min and max set to 3. Motif significance was determined by comparing the number of observed motifs with those found in the randomized networks. Genes involved in the fully connected three-node subgraphs were selected and ranked based on their frequency.

## Transcription factors and gene ontology enrichment analysis

Maize TF and GO annotations were downloaded from GRASSIUS and GOMAP[65], respectively. The enrichment analysis was conducted with the Bioconductor package clusterProfiler[66] using the function enricher() with default parameters and cut-off of $q = 0.1$, where $q$ is the $p$ value adjusted for false discovery rate using the Benjamini−Hochberg correction. GO annotations were downloaded from the database QuickGO.

## Germplasm selection and genotype data

A training set of 281 genotypes from the Goodman−Buckler diversity panel[37] was used to predict upper leaf angle (LA), tassel branch

number (TBN), ear row number (ERN), and their corresponding principal components (PhPC1, PhPC2, PhPC3) in the Ames inbred diversity panel (a.k.a. the North Central Regional Plant Introduction Station (NCRPIS) panel)[38]. First, using publicly available multi-location phenotypic data for the Goodman–Buckler panel[19,41], we fit a linear model with environment and genotype as fixed effects from which we obtained the best linear unbiased predictions (BLUPs) for each individual. The PCs of the three phenotype BLUPs, and all further phenotypic (Ph)PCs, were produced using the R function prcomp(). Data were centered and scaled before PhPC analysis. The genotype BLUPs of Goodman–Buckler panel phenotypes and PhPCs were used to train a genomic best linear unbiased prediction (GBLUP)[67] model that obtained predicted genomic estimated breeding values (GEBVs) for 2534 Ames panel inbreds. Models for BLUPs, GBLUPs, and GEBVs were conducted in R with the package ASReml-R[68]. Kinship matrices for GBLUP were produced as described in the "Heritability" section below.

Genotypic data for the two diversity panels were downloaded from Panzea (www.panzea.org) and filtered for indels and non-biallelic markers. Missing data were imputed with the nearest neighbor method where distance is defined as linkage disequilibrium between two SNPs[69]. SNPs with low minor allele frequency were filtered using a 0.01 cut-off. Prior to analysis, genotypes used in this study were converted to AGPv4 coordinates using the tool CrossMap v0.3.7[70] with the chain file from Gramene release 61[71].

Nucleotide diversity, the average pairwise difference between all pairs of genotypes[72], was measured for each SNP using all genotypes for which GBS data were available. Marker filtering and nucleotide diversity calculations were achieved using VCFtools[73].

## Phenotypic data collections
Phenotypic data were collected at the University of Illinois, Urbana Champaign over 3 years (2018–2020). Each year, 425 Ames panel lines were randomly selected from across the distribution of the predicted PhPC1 values and planted. In addition, 75 lines from the Goodman–Buckler panel were planted (same lines each year) to ensure consistency across years. Lines were planted as single-row plots in mid-May each year; 28,000 seeds per acre with 30-inch row spacing. Field design was a randomized complete block design with two replicate blocks per year. Phenotypic observations were conducted during the first week of August after the majority of genotypes had flowered. Measurements of LA were taken from the leaf immediately above the uppermost ear. If no ear was present, we selected the fifth leaf below the flag leaf. Only plants with emerged tassels were phenotyped. Angle was measured from beneath the leaf from the horizontal axis, i.e., the stalk, to the midrib. An angle of 90º would indicate an entirely upright leaf, and an angle of 0º would be perpendicular to the stalk. TBN was determined by counting every branch originating from the tassel rachis. For each genotype, three representative plants per plot were measured.

For each trait, we applied the mixed linear model to obtain BLUPs for each genotype:

$$Y_{ijk} = \mu + G_i + E_j + Bk_{(j)} + \epsilon_{ijks} \tag{4}$$

where, the phenotype ($Y$) is explained by the $i^{th}$ genotype ($G$) observed in the $k^{th}$ block ($B$) nested in the $j^{th}$ year ($E$). Individual plants within a plot are considered subsamples ($s$).

After removing outliers, BLUPs for 1064 and 1072 genotypes for LA and TBN, respectively, were obtained.

## Heritability
Prior to estimating heritability ($\widehat{h}^2$), SNP partitions were pruned with Plink 1.9[74] to remove markers in LD of 0.7 or greater within a 50 bp window. The window was shifted 5 bps and pruning was repeated. For a given pruned SNP partition, a kinship matrix K was produced with the following model,

$$K = \frac{XX'}{n_p} \tag{5}$$

where $X$ is the normalized SNP matrix, $X'$ is its transpose, and $n_p$ is the number or SNPs in the given partition. Narrow-sense heritability ($\widehat{h}^2$) is estimated as $\frac{\widehat{\sigma_G}^2}{\widehat{\sigma_P}^2}$ where $\widehat{\sigma_G}^2$ is the additive genetic variance estimate and $\widehat{\sigma_P}^2$ is the total phenotypic variance when fitted using REML[75,76]. We generated null distributions by estimating $\widehat{h}^2$ for 1000 random gene sets using the SNPs found within their proximal regulatory region ($\pm2$ kb from TSS and TTS). Random sets had an equal number of genes compared to the partition being tested. Genes in a given partition were removed from the entire genome-wide set before random selection. The software LDAK[77] was used to produce kinship matrices and estimate $\widehat{h}^2$.

## Marker subsetting based on network analyses
Genomic coordinates of gene sets derived from network analysis approaches (those from select co-expression modules and those most highly connected in three-node subgraphs) were retrieved and imported in R. We used the Bioconductor package GenomicRanges[78] to select makers within the genomic windows defined as $\pm2$ kb from the TSS and the TTS of the co-expressed genes. Marker coordinates were intersected with the gene coordinates using the function findOverlaps() with the options type = within, ignore.strand = T.

## Genome-wide association studies
We conducted single- and multi-trait association studies for LA and TBN. Single trait associations were conducted using Bayesian-information and linkage-disequilibrium iteratively nested keyway (BLINK)[79] conducted in GAPIT[80]. Before testing SNPs, the Bayesian-information criteria (BIC)[81] was used to select the models with the optimal number of PCs using the Ames panel genome-wide pruned SNP dataset.

To test for pleiotropic associations, we utilized two approaches: (i) BLINK, where the response variable was either the first or second PhPC of LA and TBN (PhPC1, PhPC2); and (ii) multivariate extension of MLM (mvMLM), where the response was an $n$-by-$t$ matrix with $n$ being the number of observations and $t$ the number of traits[82] conducted in GEMMA[83]. For mvMLM, we conducted a leave-one-chromosome-out kinship approach[84]. Since kinship was chromosome-specific, the BIC optimal number of PCs was considered on a chromosome-specific basis. Multiple testing correction was conducted based on the number of SNPs in a given partition using the Benjamini & Hochberg false discovery rate (FDR) procedure[85]. Genes with SNPs with an FDR-adjusted $p$-value below 0.2 were considered for further analysis.

## Candidate gene association analyses
We performed single-trait association studies in the maize NAM using publicly available LA and TBN phenotypes[19,41]. RIL families with segregating markers within ±2 kb of the genic region of *ereb184* and *zhd21* were only considered and tested separately to identify NAM founder genotypes that may have causal mutations. The NAM partly overcomes the issue of signals being correlated with relatedness, as is typically found in diversity panels[86]. Thus, a generalized linear model (GLM) was used to test marker-phenotype associations. Bonferroni was used to control type I error rate at $\alpha = 0.05$.

Whole-genome-sequencing data for a set of 424 lines[48] were used to conduct a candidate gene association analysis based on a multi-trait

GLM model at the *ereb184* locus including the 5 kb upstream region harboring the SV using the R package ASReml-R[68].

## Statistical analysis of *ereb184* genetic interactions

We tested for epistatic interactions by modifying the unified MLM as follows:

$$Y = Q\gamma + S_1\alpha_1 + S_2\alpha_2 + S_1S_2\beta + Z\mu + \varepsilon \qquad (6)$$

where $Y$ is $n$ vector of phenotype BLUPs with $n$ being the number of observations; $Q$ is the $n$-by-$(p+1)$ incidence matrix corresponding to the intercept, as well as $p$ fixed effect covariates (i.e., principal components) accounting for subpopulation structure; $S_1$ is an $n$-by-1 incidence vector for the peak-associated SNP from *ereb184*; $S_2$ is an $n$-by-1 incidence vector for the testing SNP and $S_1S_2$ their interaction; $\alpha_1$ is the additive effect of the peak-associated SNP; $\alpha_2$ is the additive effect of the testing SNP; $\beta$ is the additive x additive epistatic effect between the peak-associated SNP and the testing SNP; $Z$ is an $n$-by-$n$ incidence matrix relating u to $Y$; $\mu \sim N(0, 2K\sigma_G^2)$; and $\varepsilon \sim MVN(0, I\sigma_\varepsilon^2)$ is the residual error with variance with $I$ being the identity matrix and $\sigma_\varepsilon^2$ the residual variance. The peak-associated SNP and the testing SNP are treated as fixed effects. The model was performed so that $S_2$ was an SNP assigned to a motif gene and run for each SNP in the motif gene partition including those assigned to *ereb184* that were not $S_1$. The model was run in ASReml-R [68].

## Sorghum LA association study

We retrieved sorghum orthologs based on ref. [87], which identified 11,000 sorghum-maize syntenic orthologs. Of the 200 maize TFs within our top-ranked network motif connectedness, we identified 146 sorghum-maize syntenic orthologs. Sorghum BTx623 reference (version 3) gene coordinates were retrieved from the GFF (Phytozome v12.1[88]). Sorghum has larger LD blocks than maize[89]; therefore, for each sorghum-maize syntenic ortholog, we extended the gene coordinates of ±10 kb from TSS and TTS, respectively using the R package GenomicRanges and the function start() and end(). These sorghum orthologs with extended coordinates were used to subset proximal makers using the Bioconductor function findOverlaps() with the option type = "within" and ignore.strand = T.

Sorghum LA phenotype data were previously collected for 296 individuals from the Sorghum association panel (SAP)[43] from the leaf below the flag leaf[44]. SAP GBS data[90] were filtered at MAF 0.05. All analyses in sorghum were conducted according to the methods described above for maize.

## *ereb184* SV analysis in the Goodman–Buckler panel

We retrieved existing sequencing data[49] from the Goodman–Buckler panel aligned to maize B73 AGPv4. Using Samtools v1.9, we extracted the total number of reads flagged as Q30 aligned to the genomic region defined as 1:286721317-286726162. To identify local differences within the defined region, we divided it into five equal-sized bins (969 bp) and recorded the number of aligned reads for each bin. The presence/absence of the structural variation (SV) was calculated as the ratio between the number of mapped reads in the bin and the total number of reads mapped to the entire region.

## Analysis of *zhd* UniformMu insertion lines

The *zhd1* (Mu ID: *mu1022277*, AGPv4 coordinates: Chr4:12135894-12135902), *zhd21-1* (Mu ID: mu1056071, AGPv4 Chr3:137588503.137590511), and *zhd21-2* (Mu ID: mu1018735, AGPv4 coordinates: Chr3:137588828-137590836) alleles were isolated in the W22 inbred line carrying exonic Mutator (Mu) transposon insertions as part of the UniformMu transposon collection[91]. *Zhd* alleles were backcrossed into the W22 inbred line for at least two generations. Individual homozygous mutant alleles were grown at the Danforth Center Field Research Site during the 2023 season, utilizing a two-row design with a spacing of 2.5 feet between rows and 3 feet between ranges. Each genotype was represented by 40 plants. TBN and LA measurements were collected as described above.

Segregating populations of *zhd1* and *zhd21-2* mutant alleles were generated by genetic crosses. Maize plants were grown at North Carolina State Method Road Greenhouse under 16 h of supplemental light/8 h dark, and relative temperatures of 29.4 °C day and 23.9 °C night, and 12-inch pots in Metro-Mix 830-F3B (SunGro Horticulture). Genomic DNA was isolated from leaf tissue, and gene-specific and transposon multiplex primer PCR was performed under standard conditions with 2X GoTaq Green Master Mix (Promega) with 5% DMSO (v/v). The insertions were confirmed, as was co-segregation of the phenotypes with the insertions, by PCR analysis using primers at the *zhd1* (CTCCTGGGGTTTGCAATTGC; GTGTGCATCATGTTCAGCGG) and *zhd21* (TTGTTGCAGCGTGAGACAGG; AGAAATCCATGGA-GACTCCGC) loci in combination with Mu-TIR primer (AGA-GAAGCCAACGCCAWCGCCTCYATTTCGTC). Tassel and leaf phenotypic data were collected from a population that segregated 1:1:1:1 for the following genotypes: *zhd1/+; zhd21-2/+, zhd1/zhd1; zhd21-2/+, zhd1/+; zhd21-2/zhd21-2, zhd1/zhd1; zhd21-2/zhd21-2*. Phenotypic data were not collected on double mutant plants due to delayed maturation. The blade/sheath boundary from mature leaves was scanned on each side with an Epson V600 flatbed scanner. Blade angle was quantified from scanned images of leaves with the angle tool and "measure" function in ImageJ.

## ATAC-seq libraries and data analyses

Frozen, ground shoot apex 2 tissue (-0.25 g) was resuspended in 4 mL of 1x nuclei isolation buffer (16 mM HEPES; pH8, 200 mM sucrose, 0.8 mM MgCl$_2$, 4 mM KCl, 32% Glycerol, 0.25% Triton X-100, 1x complete protease inhibitor, 0.1% 2-ME, 0.1 mM PMSF), shaken at 4 °C for 20 min and filtered through two sheets of miracloth. Nuclei were centrifuged at 1000×g at 4 °C for 15 min, resuspended in 800, 400, and finally 100 μL 1x TB (tagmentation buffer; 10 mM Tris Base, 5 mM MgCl$_2$, 10% v/v dimethylformamide), and centrifuged for 5 min at 1000×g at 4 °C between each resuspension. Libraries were generated using reagents from the Illumina DNA Library Prep Kit (FC-121-1031). About 2.5 μL Tn5 enzyme, 2.5 μL of 2x TB, and 20 μL of nuclei were incubated for 1 h at 37 °C. About 22 μL of H$_2$O, 2.5 μL 10% SDS, and 0.5 μL of Proteinase K were added to the reaction and incubated at 55 °C for 1 h. Tagmented libraries were purified using the Zymo clean and concentrator kit, eluting with RSB. 25 ng tagmented DNA was combined with 2.5 μL of both index primers (Illumina; FC-121-1011), 7.5 μL PCR master mix, 2.5 μL NPM and filled to 25 μL with RSB buffer. ATAC-seq libraries were PCR amplified for 11 cycles, diluted to 50 μL, and cleaned with a two-sided Ampure XP bead size selection. A 0.5:1 bead:sample ratio followed by a 1.2:1 bead:sample ratio was used to select -200–1000 bp libraries.

Paired-end 150 bp reads were sequenced on the Illumina Novaseq 6000. Raw ATAC-seq reads were trimmed as described for RNA-seq data with the additional parameters –stringency 1 -q 20. Cleaned reads were mapped to the maize AGPv4 genome using bowtie2 v2.4.5[92] with default parameters except for –very-sensitive -X 2000 –dovetail. Reads mapped to mitochondria and chloroplast were removed along with PCR duplicates and low mapping quality reads (mapping score <10). Peaks were called using MACS2 v2.1.2[93] with parameters -f BAMPE –shift -100 –extsize 200 –nomodel -B –SPMR -g 2106338117. Consensus peaks were generated using the Bioconductor package GenomicRanges[78]. BigWig files were generated using bamCoverage with the parameters –binSize 1 –normalizeUsing RPKM.

## Reporting summary

Further information on research design is available in the Nature Portfolio Reporting Summary linked to this article.

## Data availability

Raw and processed data generated in this study are available through NCBI Gene Expression Omnibus (GEO) database under accession GSE180593. This study used the following publicly available datasets: conserved non-coding sequences[49] and unmethylated regions[47] downloaded from MaizeGDB [www.maizegdb.org][94], tassel inflorescence ATAC-seq[45] (SRP241488; SRA ID: SRR10873334 and SRR10873333) [https://trace.ncbi.nlm.nih.gov/Traces/?view=study&acc=SRP241488], and leaf chromatin interaction data (Hi-C)[46] (SRP162341; SRA ID: SRR7889833 and SRR7889834) [https://trace.ncbi.nlm.nih.gov/Traces/?view=study&acc=SRP162341]. To facilitate the exploration of our data, we created a shinyAPP [https://edoardobertolini.shinyapps.io/MAIZE-TBLAR]. Source data are provided with this paper.

## Code availability

Scripts used in this study are available and archived online at Figshare [https://doi.org/10.6084/m9.figshare.27984821.v2].

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

## Acknowledgements

The authors would like to thank members of the Eveland, Lipka and Hake labs who assisted in tissue dissections and/or field phenotyping that contributed to this manuscript, namely Dr. Annis Richardson, Dr. Rajiv Parvathaneni, Dr. Samuel Leiboff, Dr. Matthew Murphy, Dr. Indrajit Kumar, Dr. Samuel Bonfim Fernandes, Ms. Zhonghui Wang, Ms. Adrianna Chepote, Dr. Yuguo Xiao, and Mr. Jake Sinkowitz. Also, a dear thanks to Dr. Todd Mockler and members of his team, Mr. Robert Lowery, Mr. Darren O'Brian, and Mr. Phil Ozersky, for their assistance in field phenotyping. We also thank Dr. Baoxing Song and Dr. M. Cinta Romay for providing the AGPv4 genome alignment BAM files for the resequencing data of the Maize 282 Association Panel. A special thank you to Kevin Reilly and his team in the Danforth Center Integrated Plant Growth Facility for their help with plant care as well as Noah Fahlgren and his team in the Danforth Data Science Core Facility with maintaining computational infrastructure used for this project. This work was funded by a National Science Foundation Plant Genome Research Project award # IOS-1733606 to A.L.E., A.E.L., and S.H.

## Author contributions

A.L.E., A.E.L., and S.H. designed the research. E.B. performed network analyses, data integration, and interpretation. B.R.R. performed genotype selection and multi-trait GWAS. M.B. and J.Y. contributed to experimental design and performed molecular and genetics experiments. E.B. and J.S. performed genetic analyses of *zhd* alleles. E.B. and A.L.E. wrote the paper with input from all authors. All authors contributed to editing the final manuscript, and read and approved the final manuscript.

## Competing interests

The authors declare no competing interests.
