## [Peer Review file · Nature Communications]

Regulatory variation controlling architectural pleiotropy in maize

Corresponding Author: Dr Andrea Eveland

Version 0:

Reviewer comments:

Reviewer #1

(Remarks to the Author)

This manuscript investigates the molecular mechanisms underlying the pleiotropy between leaf angle (LA) and tassel branch number (TBN), two important agronomic traits in maize (*Zea mays*). The study leverages a panel of maize mutants with developmental defects in TBN, LA, or both, and profiles their gene expression using RNA-seq across different developmental stages. This enables the identification of common gene networks that regulate boundaries between meristems and differentiating lateral organs, which contribute to pleiotropy between LA and TBN.

The authors integrate the expression data into tassel and leaf gene co-expression networks and identify pleiotropic factors that exhibit network rewiring in different developmental contexts. They use these networks to subset markers for multi-trait genome-wide association studies (GWAS), which identifies new loci contributing to architectural pleiotropy in maize.

In addition to identifying pleiotropic loci, this study also demonstrates the utility of context-specific gene regulatory networks for guiding GWAS in related species like sorghum. Furthermore, the authors identify new regulatory factors contributing to architectural pleiotropy in maize, such as transcription factors from the ZHD and EREB families, and uncover structural variation in the promoter of EREB184 that modulates pleiotropy between TBN and LA.

Overall, this study provides new insights into the molecular basis of architectural pleiotropy in maize, highlighting the importance of context-specific gene regulatory networks in controlling complex traits. The findings have implications for targeted manipulation of pleiotropic loci to optimize crop architecture for improved productivity and sustainability. The manuscript is well written with a clear flow of ideas and excellent presentation. However, I do have a few comments and suggestions as follows:

Majors:

1, This is really a comprehensive study, including tons of RNA-seq data from a range of mutants related to leaf angle and tassel branch number, sophisticated bioinformatics analyses like WGCNA and Network related GWAS, and functional gene validation in maize and sorghum. However, I am not convinced by the novelty of this study and somehow confused by such comprehensive study. What indeed is the novel finding in this study? Co-expression network related GWAS has been reported by Chad Myers lab more than 5 years ago. Meanwhile, the comprehensive networks underlying tassel branch number has been reported just in last year by Nature communications. Lastly, the pleiotropy of key genes, especially TFs for two related traits has been long well known. It's easy and artificial to pick up a few key genes for validation. So, I am wondering what indeed completely new thing is in this study. I would strongly recommend the authors summarize the true new stuff in just 1~2 sentences.

2, Overall, the GRN and co-expression networks identified to confer the pleiotropy between two traits lack the supports from wet experiments. RNA-seq usually manifests the bubble of transcriptome. Some wet-experiments such like DAP – seq in vitro and tsCUT&Tag in vivo could give strong confidence to the conclusion from the authors.

Minors:

1. In the Results section, the subheading "Transcriptional hierarchies" is unclear. I suggest changing it to "Gene regulatory network analysis" for clarity.
2. In the Results section, the first paragraph under the subheading "Gene network plasticity" is unclear.
3. In the Discussion section, the sentence "These results collectively suggest divergence between TBN, LA, and their pleiotropic regulation across maize subpopulations." could be more clearly expressed.

Reviewer #2

(Remarks to the Author)

Pleiotropy is widely present among crop traits and affects selection efficiency in crop breeding. In maize, leaf angle and tassel branch number or angle are key traits controlling plant architecture and they exhibit high phenotypic correlation. The underlying molecular regulatory mechanisms are largely unknown. To identify the gene networks underlying tassel branching and ligule development, this study performed comprehensive RNAseq for nine maize mutants with developmental defects in leaf angle and tassel morphology. By integrating developmental biology, network graph theory and quantitative genetics, the authors identified new factors and regulatory variation that contribute to pleiotropy in tassel and leaf architecture. Overall, this study was well designed and conducted. The dataset presented in this study provide an important resource for community. I have several important concerns that should be addressed.

- The current title does not well match with the actual content presented in the manuscript.
- It is very weird that two different alleles of *zhd1* showed different phenotypes (Figure 6). No significant phenotypic differences were observed in *zhd21-2*. The authors should perform careful sequencing to determine the mutation nature of *zhd21-1* and *zhd21-1* (and also the *zhd1* mutant allele). As *zhd21* functions as a hub, how the expression levels of other connected gene change in the *zhd21* mutant allele? More experimental evidence should be provided to support the predicted network.
- For the GWAS model, BLINK tends to give a single significant signal at each associated locus. I am wondering how the general mixed linear model performed? Are the false positives correctly controlled? The associations at *ereb184* and *zhd21* should be experimentally validated. Are they coding variation or regulatory variation? More detailed analysis should be provided for *zhd21*.
- As *ereb184* is a key TF identified in this study, but its biological function was not really tested using mutant alleles. The identified associated SNP is located in the last exon of *ereb184*, however, in the subsequent analysis, the authors jumped to the upstream regulatory region of *ereb184* and concluded that the 5-kb PAV is the causative variant. Is this PAV in high LD with the initially detected SNP from GWAS? The associations between PAV and leaf angle or tassel branch number are only marginally significant. From my point of view, the evidence of this part analysis is very weak.
- A few technical details: Why $r^2=0.7$ was used for LD filtering (line 187 in Method)? Why an FDR-adjusted P-value below 0.2 was used (line 224 in Method)? How expression time was calculated? More details should be provided.

Reviewer #3

(Remarks to the Author)

This manuscript provides an in-depth analysis of the genetic and genomic networks responsible for the pleiotropy between tassel branch number and leaf angle in maize. An increased leaf angle has been repeatedly selected in maize to optimize light capture in dense plantings, while the associated reduction in branch number is of less certain agronomic relevance. The authors explored transcriptional profiles for multiple stages of tassels and leaves across a panel of mutants that influence leaf angle and tassel branch number. This rich expression data set facilitated construction of a regulatory network for the common leaf angle/tassel branch number phenotype, and a set of gene co-expression networks. These networks were then used to select a subset of SNPs likely to be associated with leaf angle and tassel branch number for increased statistical power in a GWAS analysis. The GWAS identified *ereb184* and *zhd21* as likely regulators of the pleiotropy in a large diversity panel. Functional validation of *zhd21* and a *zhd1*, a related TF also implicated in the leaf angle/branch number pleiotropy, was performed with mutator insertions which a clear branch number and leaf angle phenotype for *zhd21*, and a complex epistatic interaction between *zhd21* and *zhd1*. Functional validation of *ereb184* was performed through a naturally present promoter variant which localized to a likely regulatory region. Surprisingly, lines containing this variant had an increase in branch number and more upright leaves, reversing the typical pleiotropy of these linked phenotypes.

Overall, I found this an impressive study. Considering the complexity of the dataset and analyses, the manuscript is concise and well written. The figures nicely summarize the data. I was particularly impressed with the diverse complementary approaches that were used to home in on novel regulators of the branch number/leaf angle connection in maize. Ultimately, the approach proved effective as they were able to validate a role for two novel regulators of this phenotype including a natural regulatory variant that reverses the typical pleiotropy. This functional confirmation validates the extensive effort and expense of the network analysis when combined with a GWAS. In my view, this study provides a compelling model of how to expand our list of candidate genes and breeding targets for developmental traits that can be more broadly employed in maize and beyond.

I have a few mostly minor comments that I believe can easily be addressed in a minor revision.

1. The claims about evolutionary constraint, pleiotropy and developmental plasticity in lines 74-76 are interesting, but I imagine somewhat contested as well. Are there some specific citations to support these claims?
2. Is there some way to indicate which stage (1-5) was sampled for each of the replicates in Figure 1c? Not critical, but could make the figure a bit more informative.
3. I believe previous work referred to *ereb184* as *ZmANT1*, perhaps that is preferable in this case since there are so many *ereb*'s and the orthologous *ANT1* has a well described function in *Arabidopsis*?
4. Is the direction of the phenotype (i.e. reduced TBN and LA) for *zhd21-1* expected given any expression changes in other

mutants your profiled? I would expect zhd21 levels to be reduced in at least some of the mutants (lg1, lg2, fun).

5. line 342-344 "Since we hypothesize some degree of functional redundancy among zhd genes, lack of phenotypes for the other alleles was not surprising." This reasoning would apply to zhd1, but not to zhd21-2 since the other allele of zhd21 had a significant phenotype. Is there something about the location of the mu insertion in zhd21-2, or its effect on expression levels that explains the lack of phenotype in this allele?

6. The interaction of zhd1 and zhd21 was unusual and intriguing. Is there some reason the double mutant (homozygous at both loci) was not included?

7. The nature of the structural variant in ereb184 was not clearly described. Is this simply an indel? That seems to be the case, but "structural variant" could imply other rearrangements. Perhaps this could be shown graphically somehow in a figure or supplement?

Version 1:

Reviewer comments:

Reviewer #1

(Remarks to the Author)

The authors have done a very good job, which fully addressed all my concerns/questions in the previous manuscript. Thank you very much! This study also provides a user-friendly interface for the exploration of the raw and analyzed data, a great biological resource for the community. I will be very happy to read such good paper in Nature Communications.

Reviewer #2

(Remarks to the Author)

The authors have adequately addressed my previous questions. I do not have further questions.

Reviewer #3

(Remarks to the Author)

The revised manuscript has considered and addressed all of my concerns. This is a really nice study and the authors have improved it significantly.

Response to Reviewers

We were delighted at the level of reviewer interest and appreciation of the work. We found the reviewers' comments especially useful for clarifying certain aspects of the manuscript and highlighting details that may not have been clear to readers. Notably, several of the comments and suggestions prompted us to conduct additional in-depth analyses that we believe greatly strengthened our manuscript and added important supporting information for readers. Based on these analyses and other requests from reviewers, we have included an additional five supplementary figures. Our analyses provided intuitive visuals for examining the structural variation in the promoter of *ereb184* across maize diversity and validated our SNP-trait associations for the two transcription factors in maize Nested Association Mapping RIL families. Some of our analyses were enabled due to data sets that became available in the meantime, including DAP-seq data from maize (Galli et al., 2024; doi:10.1101/2024.05.31.596834) and a refined and unified marker set (Andorf et al., 2024; doi:10.1101/2024.04.30.591904). We further created maize-TBLAR, an open-access web tool powered by an R Shiny app as an interactive component of the manuscript designed for readers to explore our large transcriptional dataset through a variety of features.

We believe we addressed all of the reviewers' comments and concerns. Specific responses and adjustments to the manuscript are noted below. Specific changes in the revised manuscript file are colored in blue text.

Response to Reviewer 1:

This manuscript investigates the molecular mechanisms underlying the pleiotropy between leaf angle (LA) and tassel branch number (TBN), two important agronomic traits in maize (*Zea mays*). The study leverages a panel of maize mutants with developmental defects in TBN, LA, or both, and profiles their gene expression using RNA-seq across different developmental stages. This enables the identification of common gene networks that regulate boundaries between meristems and differentiating lateral organs, which contribute to pleiotropy between LA and TBN.

The authors integrate the expression data into tassel and leaf gene co-expression networks and identify pleiotropic factors that exhibit network rewiring in different developmental contexts. They use these networks to subset markers for multi-trait genome-wide association studies (GWAS), which identifies new loci contributing to architectural pleiotropy in maize.

In addition to identifying pleiotropic loci, this study also demonstrates the utility of context-specific gene regulatory networks for guiding GWAS in related species like sorghum. Furthermore, the authors identify new regulatory factors contributing to architectural pleiotropy in maize, such as transcription factors from the ZHD and EREB families, and uncover structural variation in the promoter of *EREB184* that modulates pleiotropy between TBN and LA.

Overall, this study provides new insights into the molecular basis of architectural pleiotropy in maize, highlighting the importance of context-specific gene regulatory

networks in controlling complex traits. The findings have implications for targeted manipulation of pleiotropic loci to optimize crop architecture for improved productivity and sustainability. The manuscript is well written with a clear flow of ideas and excellent presentation. However, I do have a few comments and suggestions as follows:

Majors:

1, This is really a comprehensive study, including tons of RNA-seq data from a range of mutants related to leaf angle and tassel branch number, sophisticated bioinformatics analyses like WGCNA and Network related GWAS, and functional gene validation in maize and sorghum.

However, I am not convinced by the novelty of this study and somehow confused by such comprehensive study. What indeed is the novel founding in this study? Co-expression network related GWAS has been reported by Chad Myers lab more than 5 years ago. Meanwhile, the comprehensive networks underlying tassel branch number has been reported just in last year by Nature communications. Lastly, the pleiotropy of key genes, especially TFs for two related traits has been long well known. It's easy and artificial to pick up a few key genes for validation. So, I am wondering what indeed completely new thing is in this study. I would strongly recommend the authors summarize the true new stuff in just 1~2 sentences.

Response: We are pleased that the reviewer acknowledges the comprehensive breadth of our study, however, we respectfully disagree here about the novelty and impact. Since tassel branch number and leaf angle in maize are important agronomic traits, naturally there have been other studies looking at transcriptional networks and association studies related to these traits. Our approach to use context-specific network analysis, and in particular leveraging a series of mutants that display pleiotropy and variation in TBN and LA, to explore pleiotropy between traits is a novel contribution. Our analyses revealed the plasticity of gene networks around pleiotropic transcription factors, offering molecular insights into how these factors are repurposed in distinct developmental contexts. We expect that these datasets will be invaluable to the community for further hypothesis generation and testing.

Our approach then used biologically-informed network modules characterized by high heritability (h^2) to refine SNP marker subsets for multivariate genome-wide association studies (GWAS). This is very much distinct from previous methods used to integrate gene networks and GWAS. The mentioned prior work led by Chad Myers used gene regulatory networks to interpret GWAS results in prioritizing candidate genes. This is in stark contrast to our approach where we use transcriptional networks to define the genetic architecture of complex traits. Furthermore, our network motif analysis combined with multi-trait GWAS showed that a small subset of biologically defined markers can be used for identifying significant trait associations, including loci that we demonstrate affect the trait(s) of interest and strikingly, that can be translated directly to a closely related species

(sorghum). Therefore, this approach has far-reaching impacts for identifying novel SNP-trait associations including those of small effect, and for genomic selection.

We believe that our abstract summarizes the significant aspects of our research mentioned above. We also had included a sentence at the end of the Introduction in the previous manuscript (now lines 93-96 in the revised manuscript) that we thought summarized the novelty and contributions of the work: *“Here, we leverage this system and a novel approach that integrates developmental biology, network graph theory and quantitative genetics to identify new factors and regulatory variation that contribute to pleiotropy in tassel and leaf architecture”*. As requested by the reviewer, we have now added an additional sentence to the end of the introduction to further state the novelty and significance of the work (**lines 96-99**): *“We demonstrate that strategic integration of developmental context-specific biological data to inform reduced marker sets in association studies can enable discovery of pleiotropic loci of small effect size, which are more agronomically relevant and typically masked by large effect loci.”*.

In addition, lines 425-430 in the Discussion address this: *“By strategically integrating context-specific biological data and multivariate GWAS models that exploit maximal diversity in TBN and LA, we identified new regulatory factors contributing to architectural pleiotropy in maize. Our approach can be applied in any genetic system to disassociate pleiotropic phenotypes through manipulation of cis-regulatory components and gene network connections.”*.

2, Overall, the GRN and co-expression networks identified to confer the pleiotropy between two traits lack the supports from wet experiments. RNA-seq usually manifests the bubble of transcriptome. Some wet-experiments such like DAP-seq in vitro and tsCUT&Tag in vivo could give strong confidence to the conclusion from the authors.

Response: Thank you for the comment. While we are not clear on what is meant by “RNA-seq manifesting the bubble of transcriptome”, we do acknowledge that gene co-expression networks and gene regulatory networks including the ones presented in our manuscript are indeed predictions at the end of the day and should be treated as such. This is reiterated throughout the manuscript. The fact that we were able to identify the same two transcriptional regulators with pleiotropic effects on TBN and LA by using various subsets of markers derived from these network predictions, indicates that there is significant biological merit to our networks. We also showed an example of a subnetwork in Figure 7b that is supported by published ChIP-seq data for KN1 in maize. Several high-profile published studies have highlighted the utility of GRNs as simply a framework for targeting candidate genes underlying agronomic traits without using wet-lab experiments. Notably, Ramirez-Gonzalez et al 2018, *Science* (<https://www.science.org/doi/10.1126/science.aar6089>), used networks based on WGCNA and GENIE3 in wheat to infer coordination of homeologs during development. These algorithms, which we use in our manuscript, have extensively been shown to outperform other network methods for generating testable hypotheses. The robustness of the

networks from Ramirez-Gonzalez et al were further validated in later follow-up work by Harrington et al. in 2020 (<https://doi.org/10.1534/g3.120.401436>), which addressed specific hypotheses derived from the networks using experimental validation in wheat.

We also acknowledge that many of the high-throughput methods for determining TF-DNA binding tend to generate false-positives and false-negatives in their own capacity and so should also be treated as predictions, especially given context-dependent interactions. However, we do appreciate the suggestion to compare our network connections with these types of approaches. We were aware that a maize DAP-seq atlas was in the works when we submitted this manuscript and so did not want to repeat efforts. This resource is now reported as a pre-print on *bioRxiv*: Galli et al. (2024)

<https://www.biorxiv.org/content/10.1101/2024.05.31.596834v1.full.pdf>. The authors kindly shared their DAP-seq results for ZHD21, ZHD1, and ZHD15 ahead of publication so that we could use them to validate our GRN predictions. Overall, we found strong agreement between our predictions and these independent DAP-seq analyses as those reported by Galli et al. (2024). Our predictions for ZHD transcription factors were confirmed with an accuracy ranging from 47% to 70%. To validate these findings, we performed a rigorous statistical analysis based on a null distribution generated from 1,000 iterations. The results demonstrate that our predictions significantly exceed the 95th percentile of the null distribution, underscoring the robustness and reliability of our approach (**see figure below**). Indeed, we are excited to see this result, which further supports the robustness of

our networks. We have now added a sentence to the revised manuscript in the Methods section, **lines 615-617** “DAP-seq data for ZHD TFs supported our predictions with accuracies ranging from 47% to 70%, exceeding the 95th percentile threshold of a null distribution derived from 1,000 random permutations.” and included a citation to this recent study.

A graphical representation of the overlap between the predicted targets of ZHD21, 1, and 15, based on our context-specific gene regulatory network, and those identified using DAP-seq data (Galli et al., 2024, bioRxiv <https://www.biorxiv.org/content/10.1101/2024.05.31.596834v1.full.pdf>). The brown area represents the null distribution created from 1,000 permutations. The arrows indicate the 95th percentile of the null distribution (in green) and the prediction overlap between the gene regulatory network from leaf and tassel and the DAP-seq data (in purple).

Minors:

1. In the Results section, the subheading "Transcriptional hierarchies" is unclear. I suggest changing it to "Gene regulatory network analysis" for clarity.

Response: Thank you for this comment. While "transcriptional hierarchies" is not part of the subheading on line 179, the term is used in the first sentence of that section to refer to hierarchical levels of TF-DNA interactions. I appreciate that it may not be totally clear in this context and may not be the best way to describe what we are presenting in the results. We took this suggestion and **line 180** now reads "To determine *gene regulatory network interactions* ..".

2. In the Results section, the first paragraph under the subheading "Gene network plasticity" is unclear.

Response: I am not exactly clear of what this comment is referring to. The first paragraph under the subheading "*Gene network plasticity around pleiotropic loci in different developmental contexts*" describes the gene sets used for WGCNA. Aside from revising the first sentence based on the previous comment, this paragraph seems quite straightforward. If the reviewer is referring to the subtitle itself, gene network plasticity is frequently used to describe changes between two networks and is exactly what we are describing in this section.

3. In the Discussion section, the sentence "These results collectively suggest divergence between TBN, LA, and their pleiotropic regulation across maize subpopulations." could be more clearly expressed.

Response: Thank you very much for this suggestion. We realize that this sentence was not only not clear, but also not in the right place. We moved this sentence to the previous paragraph where the PCA in Figure 4 is referenced on **lines 261-263**, and updated this sentence for clarity to read "*These results collectively suggest that trait variation across maize subpopulations has diverged, including the pleiotropic components governing TBN and LA.*"

Response to Reviewer 2:

Pleiotropy is widely present among crop traits and affects selection efficiency in crop breeding. In maize, leaf angle and tassel branch number or angle are key traits controlling plant architecture and they exhibit high phenotypic correlation. The underlying molecular regulatory mechanisms are largely unknown. To identify the gene networks underlying tassel branching and ligule development, this study performed comprehensive RNAseq for nine maize mutants with developmental defects in leaf angle and tassel morphology. By integrating developmental biology, network graph theory and quantitative genetics, the authors identified new factors and regulatory variation that contribute to pleiotropy in tassel and leaf architecture. Overall, this study was well designed and conducted. The

dataset presented in this study provide an important resource for community. I have several important concerns that should be addressed.

- The current title does not well match with the actual content presented in the manuscript

Response: We appreciate the suggestion here but hope we can convince the reviewer that our title is in fact indicative of the work presented, as we strongly prefer to keep the current title. Architectural pleiotropy refers to the pleiotropy between architectural traits, tassel branch number and leaf angle, which is the central theme of our study. Our findings show rewiring of gene regulatory networks in the development of organs where the traits originate from, hence regulatory variation in leaf vs tassel developmental networks. We also show that expression of the gene encoding the EREB184 TF identified through our network-guided GWAS approaches is modulated by structural variation in its promoter region and this also contributes to architectural pleiotropy between the two traits. Given this, we feel the title is appropriate. Furthermore, since the preprint of this manuscript has already been cited several times, keeping the original title will allow for seamless integration of these citations.

- It is very weird that two different alleles of *zhd1* showed different phenotypes (Figure 6). No significant phenotypic differences were observed in *zhd21-2*. The authors should perform careful sequencing to determine the mutation nature of *zhd21-1* and *zhd21-1* (and also the *zhd1* mutant allele).

Response: Thank you for this comment as indeed others may have the same question and we have now added a supplemental figure in support of this (**Supplementary Fig. 12**; see below). The two independent *zhd21* alleles reported in this work have Mutator (Mu) transposon insertions located in two different genomic regions. The Mu insertion in *zhd21-1* resides in the zinc-finger domain, disrupting the dimerization region. In the original manuscript we had acknowledged this in lines 341-344 (lines 362-364 in the revised version): *“The mutation in zhd21-1 disrupts the zinc-finger domain, which we expect confers the observable phenotype. Since we hypothesize some degree of functional redundancy among zhd genes, lack of phenotypes for the other alleles was not surprising.”* The Mu insertion in *zhd21-2* is exonic but not located in any critical protein domain, such as the Zf-HD dimerization region or the homeobox domain, as shown in the graphical visualization below.

A graphical representation of the *zhd21* locus with Mutator transposon insertions depicted as red bars and protein domains shown as colored blocks. The gene structure is derived from MaizeGDB JBrowser, built on the maize reference v5.

As suggested, we performed sequencing of *zhd21-1* and *zhd21-2* alleles. A gel image below shows the amplification products using specific primer pairs noted in the Methods section of the manuscript (lines 795-798 in the revised manuscript). As expected, the two amplification reactions are specific to the allele of interest, clearly discerning the two alleles.

Agarose gel of the PCR reactions conducted on *zhd21-1* (lane 1), *zhd21-2* (lane 2), *zhd1* (lane 3), *w22 (bz1-mum9)* (lane 4)

We now provide the sequence information along with a diagrammatic representation of the locations of the two Mu insertions relative to the *zhd21* gene structure in a new **Supplementary Fig. 12**, which is referenced in **line 364** of the revised manuscript.

As *zhd21* functions as a hub, how the expression levels of other connected gene change in the *zhd21* mutant allele? More experimental evidence should be provided to support the predicted network.

Response: Further analyses related to *zhd* mutants we believe is outside of the scope of this study. We used an integrative approach to uncover ZHD21 and EREB184 as transcriptional regulators underlying pleiotropic interactions between TBN and LA. Our

analysis of the *zhd* mutant alleles confirmed the association with these two traits, providing support for our approach. Further studies on this gene family and the stacking of mutant alleles will provide more insights into the mechanisms by which these TFs fine-tune pleiotropy between these traits. Regarding additional experimental evidence to support network predictions, please see response to Reviewer 1 above and the overlap of ZHD network connections with those from DAP-seq experiments in maize.

- For the GWAS model, BLINK tends to give a single significant signal at each associated locus. I am wondering how the general mixed linear model performed? Are the false positives correctly controlled?

Response: The method Bayesian-information and linkage-disequilibrium iteratively nested keyway (BLINK) was specifically designed to minimize false positives while enhancing power (i.e., increasing true positives). The model runs iteratively, filtering for significant variants in linkage, which typically results in a single significant SNP tagging each associated region. In contrast, the generalized linear model (GLM) showed greater p -value inflation, likely due to insufficient control of population structure - a common challenge in large diversity panels like ours.

We appreciate this point, and we have now added **Supplementary Fig. 9** (referenced at **line 306**) with the Quantile-Quantile plots representing the distribution of observed versus expected $-\log_{10} P$ -values from the multivariate GWAS conducted on TBN and LA using subsets of markers within 2 kb proximity of genes in the six co-expression modules with significant h^2 (A) and in the top 200 interconnected TFs within three-node motifs (B)

The associations at *ereb184* and *zhd21* should be experimentally validated. Are they coding variation or regulatory variation? More detailed analysis should be provided for *zhd21*.

Response: The SNP-trait associations that we identified were in the coding regions of these two genes (This was stated in the previous version of the manuscript as well as the revised version at lines 308-310). As we discuss below, likely these are not the causal SNPs.

To further validate the associations at *ereb184* and *zhd21*, we performed candidate gene association using the Maize Nested Association Mapping (NAM) RIL families with publicly available LA and TBN phenotypes from Tian et al. (2011) and Brown et al. (2011), respectively. These association analyses identified peak SNP-trait associations for both traits in several of the biparental NAM RIL families spanning the entire gene locus region of both genes. This result not only supports associations of these genes with variation in the traits of interest, but visualizing the associations with SNP markers across the gene space also indicates that we cannot necessarily pinpoint the causal SNP - this supports our response to the next comment.

This is now stated in the revised manuscript at **lines 314-318**: “To further validate the associations at *ereb184* and *zhd21*, we performed candidate gene association analysis using the maize Nested Association Mapping (NAM) recombinant inbred line (RIL) families along with their publicly available phenotype data for LA and TBN^{19,41}. This analysis showed peak SNP-trait associations in several NAM families (Supplementary Fig. 10).”

We also included the analysis in the Methods section (**lines 726-736**) and in a new **Supplementary Fig. 10**.

- As *ereb184* is a key TF identified in this study, but its biological function was not really tested using mutant alleles. The identified associated SNP is located in the last exon of *ereb184*, however, in the subsequent analysis, the authors jumped to the upstream regulatory region of *ereb184* and concluded that the 5-kb PAV is the causative variant. Is this PAV in high LD with the initially detected SNP from GWAS? The associations between PAV and leaf angle or tassel branch number are only marginally significant. From my point of view, the evidence of this part analysis is very weak.

Response: We appreciate this comment because in addressing this, we added some new analyses that provided informative results and visuals in new Supplementary Figures, and we even uncovered rare variants that could potentially be causal!

Firstly, based on expression data from the NAM, we showed a significant correlation of presence/absence of the SV in the regulatory region of *ereb184* with its expression level (this is shown in Figure 7d).

In our association analyses, we used genotyping-by-sequencing (GBS) data, which are known to have varying levels of coverage throughout the genome. Due to the relative sparseness of the marker set, the associated SNP detected in the gene model cannot be considered the causal QTN; instead, it serves as a good proxy linking the genomic region to the phenotypic variation observed in this study. To support our results, we performed additional gene association validations using whole-genome sequencing (WGS) markers from Grzybowski et al. (2023) <https://onlinelibrary.wiley.com/doi/10.1111/tpj.16123> for a set of 424 maize lines. This new multivariate association analysis of *ereb184* highlighted the association of three additional significant SNPs with LA and TBN, including one within the structural variant (SV) ($-\log_{10} p\text{-value} = 7.24$). This is now included in the revised manuscript at **lines 402-406**: “The marker identified through our network-guided GWAS in the genic region of *ereb184* was in linkage disequilibrium (LD) with the SV (Supplementary Fig. 13). This result was supported by a multivariate candidate gene association analysis at the *ereb184* locus for TBN and LA using whole-genome SNPs⁴⁸, which also identified a peak-associated SNP in the SV ($-\log_{10} P\text{-value} = 7.24$).”

The new **Supplementary Fig. 13** that we added includes these new results and the LD plot spanning the SV and the gene locus, showing moderate to high LD, as well as the original network-guided GWAS SNP.

Our analysis in Figure 7 that tested whether the SV contributes to trait variation in LA and TBN was conducted independently of other loci; therefore, multiple testing corrections were not applicable. A nominal significance threshold of 0.05 was deemed reasonable for this targeted analysis. It is important to note that several factors influence the nominal p-value. We used the largest resequencing dataset available to date for a maize diversity panel ($n = 231$). Furthermore, LA was measured in 5-degree increments, which reduces variation within each increment. In contrast, TBN as a numerical variable retains its continuous nature and may therefore capture more subtle variations. This difference in variable type (discrete vs. continuous) could partly explain the lower statistical significance observed for LA compared to TBN.

To further address this comment, we selected 49 inbred lines based on genetic similarity (kinship) with the NAM founders and plotted their sequencing coverage that we used to call the PAV of the SV across the Goodman-Buckler panel. These analyses are now shown in a new **Supplementary Figure 14** (referenced on **line 411**). As shown in the new Supplementary Fig. 14A, our method clearly detected PAV. We confirmed these sequencing data with PCR using primers targeting the 3' region of the SV and the 5' UTR region of *ereb184*.

The association between the PAV and LA or TBN for this subset of 49 maize lines is reported in Supplementary Fig. 14B, and the P-values are reduced (Wilcoxon, $p = 0.02$ and $p = 0.04$ for LA and TBN, respectively).

- A few technical details: Why $r^2=0.7$ was used for LD filtering (line 187 in Method)? Why an FDR-adjusted P-value below 0.2 was used (line 224 in Method)? How expression time was calculated? More details should be provided.

Response: We removed single nucleotide polymorphisms (SNPs) that exhibited high linkage disequilibrium (LD), using an LD threshold of 0.7 within a 50-base-pair (bp) window. This approach was taken to reduce redundancy in the SNP dataset when estimating kinship. By pruning to remove markers in high LD (0.7), we retained a set of representative markers that maximizes the likelihood of tagging potential causal genetic variants while minimizing redundancy.

We acknowledge the inherent conservativeness of genome-wide association studies (GWAS) and are mindful of the risk of overlooking biologically significant associations due to smaller effect sizes falling below traditional thresholds of statistical significance. To address this, we applied an adjusted false discovery rate (FDR) of < 0.2 to filter markers for downstream analysis. This slightly less stringent cutoff was chosen intentionally, as our marker set was pre-selected based on gene networks, allowing us to capture a broader range of potential associations. In GWAS, the choice of FDR cutoff is typically left to the researcher as a tool for refining the list of genes for further exploration. By adopting a more permissive threshold, we aimed to retain markers that may exhibit subtle but potentially meaningful associations that could be missed under stricter criteria.

The expression time analysis was described in the submitted Methods of the original manuscript (now lines 572-576). *“Expression time (ET) was calculated using a smooth spline regression model 5 with the R function bs(). We fitted a b-spline (3-knot with three degrees of freedom) modeled on the first and second PC of the 500 most dynamically expressed genes across normal tassel development. Data points from mutant backgrounds were classified based on their location on the spline in relation to this model.”*

Response to Reviewer 3:

This manuscript provides an in-depth analysis of the genetic and genomic networks responsible for the pleiotropy between tassel branch number and leaf angle in maize. An increased leaf angle has been repeatedly selected in maize to optimize light capture in dense plantings, while the associated reduction in branch number is of less certain agronomic relevance. The authors explored transcriptional profiles for multiple stages of tassels and leaves across a panel of mutants that influence leaf angle and tassel branch number. This rich expression data set facilitated construction of a regulatory network for the common leaf angle/tassel branch number phenotype, and a set of gene co-expression networks. These networks were then used to select a subset of SNPs likely to be associated with leaf angle and tassel branch number for increased statistical power in a GWAS analysis. The GWAS identified *ereb184* and *zhd21* as likely regulators of the pleiotropy in a large diversity panel. Functional validation of *zhd21* and a *zhd1*, a related TF also implicated in the leaf angle/branch number pleiotropy, was performed with mutator insertions which a clear branch number and leaf angle phenotype for *zhd21*, and a complex epistatic interaction between *zhd21* and *zhd1*. Functional validation of *ereb184* was performed through a naturally present promoter variant which localized to a likely regulatory region. Surprisingly, lines containing this variant had an increase in branch number and more upright leaves, reversing the typical pleiotropy of these linked phenotypes.

Overall, I found this an impressive study. Considering the complexity of the dataset and analyses, the manuscript is concise and well written. The figures nicely summarize the data. I was particularly impressed with the diverse complementary approaches that were used to home in on novel regulators of the branch number/leaf angle connection in maize. Ultimately, the approach proved effective as they were able to validate a role for two novel regulators of this phenotype including a natural regulatory variant that reverses the typical pleiotropy. This functional confirmation validates the extensive effort and expense of the network analysis when combined with a GWAS. In my view, this study provides a compelling model of how to expand our list of candidate genes and breeding targets for developmental traits that can be more broadly employed in maize and beyond.

I have a few mostly minor comments that I believe can easily be addressed in a minor revision.

1. The claims about evolutionary constraint, pleiotropy and developmental plasticity in lines 74-76 are interesting, but I imagine somewhat contested as well. Are there some specific citations to support these claims?

Response: Thank you for noting this and we agree. We added a few reviews as references here: Auge et al 2019 is a review describing effects of genetic pleiotropy on trait variation in flowering time, Hill et al. 2020 discusses molecular and evolutionary processes underlying gene expression variation, and Pavličev and Cheverud 2015 discusses context dependency and the evolution of pleiotropy. We also changed “*major cause of evolutionary constraint*” to “*significant cause of evolutionary constraint*” on **line 86**.

2. Is there some way to indicate which stage (1-5) was sampled for each of the replicates in Figure 1c? Not critical, but could make the figure a bit more informative.

Response: Thank you for this suggestion! We have updated **Figure 1c** to indicate immature tassel stages 1–5, and we agree this makes the figure more intuitive.

3. I believe previous work referred to ereb184 as ZmANT1, perhaps that is preferable in this case since there are so many ereb’s and the orthologous ANT1 has a well described function in Arabidopsis?

Response: We appreciate the suggestion here. However, the maize TFome nomenclature established by Yilmaz et al. (2009) <https://doi.org/10.1104/pp.108.128579> is widely recognized as a standard in the maize research community, including as standards on MaizeGDB, and we have chosen to adhere to this system in our manuscript. Using the designation 'ereb###' provides a clear and unambiguous reference to the gene in question.

4. is the direction of the phenotype (i.e. reduced TBN and LA) for zhd21-1 expected given any expression changes in other mutants your profiled? I would expect zhd21 levels to be reduced in at least some of the mutants (lg1, lg2, fun).

Response: Based on our expression data in the mutants, *zhd21* was down-regulated in developing leaf/ligule samples in several mutants with upright LA phenotypes (*lg2*, *fun*, and *bri1*) but up-regulated in *wab1R*, consistent with *zhd21-1* mutants displaying upright LA. We also found that *zhd21* was up-regulated in tassel primordia of *ra1* and *ra2* mutants, which make more tassel branches, consistent with a reduction in TBN in the *zhd21-1* mutant. While these results trend in a consistent direction, we acknowledge that the interconnected *zhd* regulatory subnetworks underlying these traits appear to be very complex and potentially redundant.

We appreciate the suggestion from the reviewer to look into the expression of *zhd21* in the mutants and we added a sentence to the Discussion **lines 462-465**: “*Notably, while zhd21 was down-regulated in lg2 developing leaf/ligule, there was no significant*

difference in its expression in *lg2* tassels, which make few to no tassel branches, potentially consistent with *zhds* regulating pleiotropic effects between these traits”.

We realized that this type of question may be common within the larger plant developmental community, so we decided to develop a **publicly accessible Shiny app**, available at <https://edoardobertolini.shinyapps.io/MAIZE-TBLAR>, to make it easy to explore our large dataset. This is now referenced in the Methods (lines 1122-1123). Below are screenshots of a table and graph views from our interactive app, where users can explore the differential expression of specific genes in the mutants we profiled.

AGPv4	Log2FC	FDR	Comparison	maizeGDB symbol
Zm00001d041780	-1.10262077	1.1E-05	bri1 - Shoot Apex 2	zhd21
Zm00001d041780	-0.71185125	0.01213	fun - Shoot Apex 1	zhd21
Zm00001d041780	-0.70535417	0.01915	lg2 - Shoot Apex 2	zhd21
Zm00001d041780	0.88520878	0.00032	wab1R - Shoot Apex 1	zhd21
Zm00001d041780	0.54844244	0.0467	ra2 - stage1	zhd21
Zm00001d041780	0.61307739	0.0138	ra1 - stage1	zhd21
Zm00001d041780	0.63785	0.00558	ra2 - stage2	zhd21

Differential expression (DE) data of *zhd21* across different comparisons in the profiled mutants and tissues. The table reports the Log₂ Fold Change (Log₂FC) of DE (mutant vs. normal, in the same developmental condition).

Expression levels (TPM) of *zhd21* (Zm00001d041780) across the profiled mutants in tassel stages.

Expression levels (TPM) of *zhd21* (Zm00001d041780) across the profiled mutants in shoot apices.

5. line 342-344 “Since we hypothesize some degree of functional redundancy among *zhd* genes, lack of phenotypes for the other alleles was not surprising.” This reasoning would apply to *zhd1*, but not to *zhd21-2* since the other allele of *zhd21* had a significant phenotype. Is there something about the location of the mu insertion in *zhd21-2*, or its effect on expression levels that explains the lack of phenotype in this allele?

Response: Yes! We addressed this point in a response to a similar question from Reviewer 2. Please see that response above and the associated **Supplementary Fig. 12**, which was added for reader clarity.

6. The interaction of *zhd1* and *zhd21* was unusual and intriguing. Is there some reason the double mutant (homozygous at both loci) was not included?

Response: We appreciate this comment. We did mention in the Methods section that “Phenotypic data were not collected on double mutant plants due to delayed maturation”. The double mutant plants between *zhd21* and *zhd1* have more complex phenotypes including very late and abnormal flowering, making it unclear on how to developmentally stage the plants for accurate phenotypic comparisons. We feel that a separate study on the genetic and mechanistic relations between these *zhds* would be warranted and it is outside the scope of this study to branch into an intensive developmental analysis of stacked *zhd* mutants. Co-author Strable is currently working on a more comprehensive study of the role of these *zhds* in maize development.

7. The nature of the structural variant in *ereb184* was not clearly described. Is this simply an indel? That seems to be the case, but “structural variant” could imply other rearrangements. Perhaps this could be shown graphically somehow in a figure or supplement?

Response: We appreciate this suggestion. In the Discussion at line 448 of the original manuscript (now lines 477-480) we did address this point by reporting the results of a blast search that revealed similarity with retrotransposons and Helitron elements. “*A blast search of the SV DNA sequence against a curated transposable element database revealed the presence of a probable LTR Gypsy retrotransposon (86% alignment identity) in the 3-prime region of the SV, as well as interspersed Helitron fragments*”.

To make this clearer for readers, we now include a graphical representation of the structural variant in the genome. To further validate our findings, we selected 49 inbred lines based on genetic similarity (kinship) with the NAM founders and plotted, as genome view, the sequencing coverage that we used to call the PAV of the SV across the Goodman-Buckler panel. This is now presented in **Supplementary Fig. 14 (line 411)**. We further confirmed this by PCR with primers targeting the 3' SV region and the 5' UTR region of *ereb184*.

Response to Reviewers

We were delighted at the level of reviewer interest and appreciation of the work. We found the reviewers' comments especially useful for clarifying certain aspects of the manuscript and highlighting details that may not have been clear to readers. Notably, several of the comments and suggestions prompted us to conduct additional in-depth analyses that we believe greatly strengthened our manuscript and added important supporting information for readers. Based on these analyses and other requests from reviewers, we have included an additional five supplementary figures. Our analyses provided intuitive visuals for examining the structural variation in the promoter of *ereb184* across maize diversity and validated our SNP-trait associations for the two transcription factors in maize Nested Association Mapping RIL families. Some of our analyses were enabled due to data sets that became available in the meantime, including DAP-seq data from maize (Galli et al., 2024; doi:10.1101/2024.05.31.596834) and a refined and unified marker set (Andorf et al., 2024; doi:10.1101/2024.04.30.591904). We further created maize-TBLAR, an open-access web tool powered by an R Shiny app as an interactive component of the manuscript designed for readers to explore our large transcriptional dataset through a variety of features.

We believe we addressed all of the reviewers' comments and concerns. Specific responses and adjustments to the manuscript are noted below. Specific changes in the revised manuscript file are colored in blue text.

Response to Reviewer 1:

This manuscript investigates the molecular mechanisms underlying the pleiotropy between leaf angle (LA) and tassel branch number (TBN), two important agronomic traits in maize (*Zea mays*). The study leverages a panel of maize mutants with developmental defects in TBN, LA, or both, and profiles their gene expression using RNA-seq across different developmental stages. This enables the identification of common gene networks that regulate boundaries between meristems and differentiating lateral organs, which contribute to pleiotropy between LA and TBN.

The authors integrate the expression data into tassel and leaf gene co-expression networks and identify pleiotropic factors that exhibit network rewiring in different developmental contexts. They use these networks to subset markers for multi-trait genome-wide association studies (GWAS), which identifies new loci contributing to architectural pleiotropy in maize.

In addition to identifying pleiotropic loci, this study also demonstrates the utility of context-specific gene regulatory networks for guiding GWAS in related species like sorghum. Furthermore, the authors identify new regulatory factors contributing to architectural pleiotropy in maize, such as transcription factors from the ZHD and EREB families, and uncover structural variation in the promoter of *EREB184* that modulates pleiotropy between TBN and LA.

Overall, this study provides new insights into the molecular basis of architectural pleiotropy in maize, highlighting the importance of context-specific gene regulatory

networks in controlling complex traits. The findings have implications for targeted manipulation of pleiotropic loci to optimize crop architecture for improved productivity and sustainability. The manuscript is well written with a clear flow of ideas and excellent presentation. However, I do have a few comments and suggestions as follows:

Majors:

1, This is really a comprehensive study, including tons of RNA-seq data from a range of mutants related to leaf angle and tassel branch number, sophisticated bioinformatics analyses like WGCNA and Network related GWAS, and functional gene validation in maize and sorghum.

However, I am not convinced by the novelty of this study and somehow confused by such comprehensive study. What indeed is the novel founding in this study? Co-expression network related GWAS has been reported by Chad Myers lab more than 5 years ago. Meanwhile, the comprehensive networks underlying tassel branch number has been reported just in last year by Nature communications. Lastly, the pleiotropy of key genes, especially TFs for two related traits has been long well known. It's easy and artificial to pick up a few key genes for validation. So, I am wondering what indeed completely new thing is in this study. I would strongly recommend the authors summarize the true new stuff in just 1~2 sentences.

Response: We are pleased that the reviewer acknowledges the comprehensive breadth of our study, however, we respectfully disagree here about the novelty and impact. Since tassel branch number and leaf angle in maize are important agronomic traits, naturally there have been other studies looking at transcriptional networks and association studies related to these traits. Our approach to use context-specific network analysis, and in particular leveraging a series of mutants that display pleiotropy and variation in TBN and LA, to explore pleiotropy between traits is a novel contribution. Our analyses revealed the plasticity of gene networks around pleiotropic transcription factors, offering molecular insights into how these factors are repurposed in distinct developmental contexts. We expect that these datasets will be invaluable to the community for further hypothesis generation and testing.

Our approach then used biologically-informed network modules characterized by high heritability (h^2) to refine SNP marker subsets for multivariate genome-wide association studies (GWAS). This is very much distinct from previous methods used to integrate gene networks and GWAS. The mentioned prior work led by Chad Myers used gene regulatory networks to interpret GWAS results in prioritizing candidate genes. This is in stark contrast to our approach where we use transcriptional networks to define the genetic architecture of complex traits. Furthermore, our network motif analysis combined with multi-trait GWAS showed that a small subset of biologically defined markers can be used for identifying significant trait associations, including loci that we demonstrate affect the trait(s) of interest and strikingly, that can be translated directly to a closely related species

(sorghum). Therefore, this approach has far-reaching impacts for identifying novel SNP-trait associations including those of small effect, and for genomic selection.

We believe that our abstract summarizes the significant aspects of our research mentioned above. We also had included a sentence at the end of the Introduction in the previous manuscript (now lines 93-96 in the revised manuscript) that we thought summarized the novelty and contributions of the work: *“Here, we leverage this system and a novel approach that integrates developmental biology, network graph theory and quantitative genetics to identify new factors and regulatory variation that contribute to pleiotropy in tassel and leaf architecture”*. As requested by the reviewer, we have now added an additional sentence to the end of the introduction to further state the novelty and significance of the work (**lines 96-99**): *“We demonstrate that strategic integration of developmental context-specific biological data to inform reduced marker sets in association studies can enable discovery of pleiotropic loci of small effect size, which are more agronomically relevant and typically masked by large effect loci.”*.

In addition, lines 425-430 in the Discussion address this: *“By strategically integrating context-specific biological data and multivariate GWAS models that exploit maximal diversity in TBN and LA, we identified new regulatory factors contributing to architectural pleiotropy in maize. Our approach can be applied in any genetic system to disassociate pleiotropic phenotypes through manipulation of cis-regulatory components and gene network connections.”*.

2, Overall, the GRN and co-expression networks identified to confer the pleiotropy between two traits lack the supports from wet experiments. RNA-seq usually manifests the bubble of transcriptome. Some wet-experiments such like DAP-seq in vitro and tsCUT&Tag in vivo could give strong confidence to the conclusion from the authors.

Response: Thank you for the comment. While we are not clear on what is meant by “RNA-seq manifesting the bubble of transcriptome”, we do acknowledge that gene co-expression networks and gene regulatory networks including the ones presented in our manuscript are indeed predictions at the end of the day and should be treated as such. This is reiterated throughout the manuscript. The fact that we were able to identify the same two transcriptional regulators with pleiotropic effects on TBN and LA by using various subsets of markers derived from these network predictions, indicates that there is significant biological merit to our networks. We also showed an example of a subnetwork in Figure 7b that is supported by published ChIP-seq data for KN1 in maize. Several high-profile published studies have highlighted the utility of GRNs as simply a framework for targeting candidate genes underlying agronomic traits without using wet-lab experiments. Notably, Ramirez-Gonzalez et al 2018, *Science* (<https://www.science.org/doi/10.1126/science.aar6089>), used networks based on WGCNA and GENIE3 in wheat to infer coordination of homeologs during development. These algorithms, which we use in our manuscript, have extensively been shown to outperform other network methods for generating testable hypotheses. The robustness of the

networks from Ramirez-Gonzalez et al were further validated in later follow-up work by Harrington et al. in 2020 (<https://doi.org/10.1534/g3.120.401436>), which addressed specific hypotheses derived from the networks using experimental validation in wheat.

We also acknowledge that many of the high-throughput methods for determining TF-DNA binding tend to generate false-positives and false-negatives in their own capacity and so should also be treated as predictions, especially given context-dependent interactions. However, we do appreciate the suggestion to compare our network connections with these types of approaches. We were aware that a maize DAP-seq atlas was in the works when we submitted this manuscript and so did not want to repeat efforts. This resource is now reported as a pre-print on *bioRxiv*: Galli et al. (2024)

<https://www.biorxiv.org/content/10.1101/2024.05.31.596834v1.full.pdf>. The authors kindly shared their DAP-seq results for ZHD21, ZHD1, and ZHD15 ahead of publication so that we could use them to validate our GRN predictions. Overall, we found strong agreement between our predictions and these independent DAP-seq analyses as those reported by Galli et al. (2024). Our predictions for ZHD transcription factors were confirmed with an accuracy ranging from 47% to 70%. To validate these findings, we performed a rigorous statistical analysis based on a null distribution generated from 1,000 iterations. The results demonstrate that our predictions significantly exceed the 95th percentile of the null distribution, underscoring the robustness and reliability of our approach (**see figure below**). Indeed, we are excited to see this result, which further supports the robustness of

our networks. We have now added a sentence to the revised manuscript in the Methods section, **lines 615-617** “DAP-seq data for ZHD TFs supported our predictions with accuracies ranging from 47% to 70%, exceeding the 95th percentile threshold of a null distribution derived from 1,000 random permutations.” and included a citation to this recent study.

A graphical representation of the overlap between the predicted targets of ZHD21, 1, and 15, based on our context-specific gene regulatory network, and those identified using DAP-seq data (Galli et al., 2024, bioRxiv <https://www.biorxiv.org/content/10.1101/2024.05.31.596834v1.full.pdf>). The brown area represents the null distribution created from 1,000 permutations. The arrows indicate the 95th percentile of the null distribution (in green) and the prediction overlap between the gene regulatory network from leaf and tassel and the DAP-seq data (in purple).

Minors:

1. In the Results section, the subheading "Transcriptional hierarchies" is unclear. I suggest changing it to "Gene regulatory network analysis" for clarity.

Response: Thank you for this comment. While "transcriptional hierarchies" is not part of the subheading on line 179, the term is used in the first sentence of that section to refer to hierarchical levels of TF-DNA interactions. I appreciate that it may not be totally clear in this context and may not be the best way to describe what we are presenting in the results. We took this suggestion and **line 180** now reads "To determine *gene regulatory network interactions* ..".

2. In the Results section, the first paragraph under the subheading "Gene network plasticity" is unclear.

Response: I am not exactly clear of what this comment is referring to. The first paragraph under the subheading "*Gene network plasticity around pleiotropic loci in different developmental contexts*" describes the gene sets used for WGCNA. Aside from revising the first sentence based on the previous comment, this paragraph seems quite straightforward. If the reviewer is referring to the subtitle itself, gene network plasticity is frequently used to describe changes between two networks and is exactly what we are describing in this section.

3. In the Discussion section, the sentence "These results collectively suggest divergence between TBN, LA, and their pleiotropic regulation across maize subpopulations." could be more clearly expressed.

Response: Thank you very much for this suggestion. We realize that this sentence was not only not clear, but also not in the right place. We moved this sentence to the previous paragraph where the PCA in Figure 4 is referenced on **lines 261-263**, and updated this sentence for clarity to read "*These results collectively suggest that trait variation across maize subpopulations has diverged, including the pleiotropic components governing TBN and LA.*"

Response to Reviewer 2:

Pleiotropy is widely present among crop traits and affects selection efficiency in crop breeding. In maize, leaf angle and tassel branch number or angle are key traits controlling plant architecture and they exhibit high phenotypic correlation. The underlying molecular regulatory mechanisms are largely unknown. To identify the gene networks underlying tassel branching and ligule development, this study performed comprehensive RNAseq for nine maize mutants with developmental defects in leaf angle and tassel morphology. By integrating developmental biology, network graph theory and quantitative genetics, the authors identified new factors and regulatory variation that contribute to pleiotropy in tassel and leaf architecture. Overall, this study was well designed and conducted. The

dataset presented in this study provide an important resource for community. I have several important concerns that should be addressed.

- The current title does not well match with the actual content presented in the manuscript

Response: We appreciate the suggestion here but hope we can convince the reviewer that our title is in fact indicative of the work presented, as we strongly prefer to keep the current title. Architectural pleiotropy refers to the pleiotropy between architectural traits, tassel branch number and leaf angle, which is the central theme of our study. Our findings show rewiring of gene regulatory networks in the development of organs where the traits originate from, hence regulatory variation in leaf vs tassel developmental networks. We also show that expression of the gene encoding the EREB184 TF identified through our network-guided GWAS approaches is modulated by structural variation in its promoter region and this also contributes to architectural pleiotropy between the two traits. Given this, we feel the title is appropriate. Furthermore, since the preprint of this manuscript has already been cited several times, keeping the original title will allow for seamless integration of these citations.

- It is very weird that two different alleles of *zhd1* showed different phenotypes (Figure 6). No significant phenotypic differences were observed in *zhd21-2*. The authors should perform careful sequencing to determine the mutation nature of *zhd21-1* and *zhd21-1* (and also the *zhd1* mutant allele).

Response: Thank you for this comment as indeed others may have the same question and we have now added a supplemental figure in support of this (**Supplementary Fig. 12**; see below). The two independent *zhd21* alleles reported in this work have Mutator (Mu) transposon insertions located in two different genomic regions. The Mu insertion in *zhd21-1* resides in the zinc-finger domain, disrupting the dimerization region. In the original manuscript we had acknowledged this in lines 341-344 (lines 362-364 in the revised version): *“The mutation in zhd21-1 disrupts the zinc-finger domain, which we expect confers the observable phenotype. Since we hypothesize some degree of functional redundancy among zhd genes, lack of phenotypes for the other alleles was not surprising.”* The Mu insertion in *zhd21-2* is exonic but not located in any critical protein domain, such as the Zf-HD dimerization region or the homeobox domain, as shown in the graphical visualization below.

A graphical representation of the *zhd21* locus with Mutator transposon insertions depicted as red bars and protein domains shown as colored blocks. The gene structure is derived from MaizeGDB JBrowser, built on the maize reference v5.

As suggested, we performed sequencing of *zhd21-1* and *zhd21-2* alleles. A gel image below shows the amplification products using specific primer pairs noted in the Methods section of the manuscript (lines 795-798 in the revised manuscript). As expected, the two amplification reactions are specific to the allele of interest, clearly discerning the two alleles.

Agarose gel of the PCR reactions conducted on *zhd21-1* (lane 1), *zhd21-2* (lane 2), *zhd1* (lane 3), *w22 (bz1-mum9)* (lane 4)

We now provide the sequence information along with a diagrammatic representation of the locations of the two Mu insertions relative to the *zhd21* gene structure in a new **Supplementary Fig. 12**, which is referenced in **line 364** of the revised manuscript.

As *zhd21* functions as a hub, how the expression levels of other connected gene change in the *zhd21* mutant allele? More experimental evidence should be provided to support the predicted network.

Response: Further analyses related to *zhd* mutants we believe is outside of the scope of this study. We used an integrative approach to uncover ZHD21 and EREB184 as transcriptional regulators underlying pleiotropic interactions between TBN and LA. Our

analysis of the *zhd* mutant alleles confirmed the association with these two traits, providing support for our approach. Further studies on this gene family and the stacking of mutant alleles will provide more insights into the mechanisms by which these TFs fine-tune pleiotropy between these traits. Regarding additional experimental evidence to support network predictions, please see response to Reviewer 1 above and the overlap of ZHD network connections with those from DAP-seq experiments in maize.

- For the GWAS model, BLINK tends to give a single significant signal at each associated locus. I am wondering how the general mixed linear model performed? Are the false positives correctly controlled?

Response: The method Bayesian-information and linkage-disequilibrium iteratively nested keyway (BLINK) was specifically designed to minimize false positives while enhancing power (i.e., increasing true positives). The model runs iteratively, filtering for significant variants in linkage, which typically results in a single significant SNP tagging each associated region. In contrast, the generalized linear model (GLM) showed greater p -value inflation, likely due to insufficient control of population structure - a common challenge in large diversity panels like ours.

We appreciate this point, and we have now added **Supplementary Fig. 9** (referenced at **line 306**) with the Quantile-Quantile plots representing the distribution of observed versus expected $-\log_{10} P$ -values from the multivariate GWAS conducted on TBN and LA using subsets of markers within 2 kb proximity of genes in the six co-expression modules with significant h^2 (A) and in the top 200 interconnected TFs within three-node motifs (B)

The associations at *ereb184* and *zhd21* should be experimentally validated. Are they coding variation or regulatory variation? More detailed analysis should be provided for *zhd21*.

Response: The SNP-trait associations that we identified were in the coding regions of these two genes (This was stated in the previous version of the manuscript as well as the revised version at lines 308-310). As we discuss below, likely these are not the causal SNPs.

To further validate the associations at *ereb184* and *zhd21*, we performed candidate gene association using the Maize Nested Association Mapping (NAM) RIL families with publicly available LA and TBN phenotypes from Tian et al. (2011) and Brown et al. (2011), respectively. These association analyses identified peak SNP-trait associations for both traits in several of the biparental NAM RIL families spanning the entire gene locus region of both genes. This result not only supports associations of these genes with variation in the traits of interest, but visualizing the associations with SNP markers across the gene space also indicates that we cannot necessarily pinpoint the causal SNP - this supports our response to the next comment.

This is now stated in the revised manuscript at **lines 314-318**: “To further validate the associations at *ereb184* and *zhd21*, we performed candidate gene association analysis using the maize Nested Association Mapping (NAM) recombinant inbred line (RIL) families along with their publicly available phenotype data for LA and TBN^{19,41}. This analysis showed peak SNP-trait associations in several NAM families (Supplementary Fig. 10).”

We also included the analysis in the Methods section (**lines 726-736**) and in a new **Supplementary Fig. 10**.

- As *ereb184* is a key TF identified in this study, but its biological function was not really tested using mutant alleles. The identified associated SNP is located in the last exon of *ereb184*, however, in the subsequent analysis, the authors jumped to the upstream regulatory region of *ereb184* and concluded that the 5-kb PAV is the causative variant. Is this PAV in high LD with the initially detected SNP from GWAS? The associations between PAV and leaf angle or tassel branch number are only marginally significant. From my point of view, the evidence of this part analysis is very weak.

Response: We appreciate this comment because in addressing this, we added some new analyses that provided informative results and visuals in new Supplementary Figures, and we even uncovered rare variants that could potentially be causal!

Firstly, based on expression data from the NAM, we showed a significant correlation of presence/absence of the SV in the regulatory region of *ereb184* with its expression level (this is shown in Figure 7d).

In our association analyses, we used genotyping-by-sequencing (GBS) data, which are known to have varying levels of coverage throughout the genome. Due to the relative sparseness of the marker set, the associated SNP detected in the gene model cannot be considered the causal QTN; instead, it serves as a good proxy linking the genomic region to the phenotypic variation observed in this study. To support our results, we performed additional gene association validations using whole-genome sequencing (WGS) markers from Grzybowski et al. (2023) <https://onlinelibrary.wiley.com/doi/10.1111/tpj.16123> for a set of 424 maize lines. This new multivariate association analysis of *ereb184* highlighted the association of three additional significant SNPs with LA and TBN, including one within the structural variant (SV) ($-\log_{10} p\text{-value} = 7.24$). This is now included in the revised manuscript at **lines 402-406**: “The marker identified through our network-guided GWAS in the genic region of *ereb184* was in linkage disequilibrium (LD) with the SV (Supplementary Fig. 13). This result was supported by a multivariate candidate gene association analysis at the *ereb184* locus for TBN and LA using whole-genome SNPs⁴⁸, which also identified a peak-associated SNP in the SV ($-\log_{10} P\text{-value} = 7.24$).”

The new **Supplementary Fig. 13** that we added includes these new results and the LD plot spanning the SV and the gene locus, showing moderate to high LD, as well as the original network-guided GWAS SNP.

Our analysis in Figure 7 that tested whether the SV contributes to trait variation in LA and TBN was conducted independently of other loci; therefore, multiple testing corrections were not applicable. A nominal significance threshold of 0.05 was deemed reasonable for this targeted analysis. It is important to note that several factors influence the nominal p-value. We used the largest resequencing dataset available to date for a maize diversity panel ($n = 231$). Furthermore, LA was measured in 5-degree increments, which reduces variation within each increment. In contrast, TBN as a numerical variable retains its continuous nature and may therefore capture more subtle variations. This difference in variable type (discrete vs. continuous) could partly explain the lower statistical significance observed for LA compared to TBN.

To further address this comment, we selected 49 inbred lines based on genetic similarity (kinship) with the NAM founders and plotted their sequencing coverage that we used to call the PAV of the SV across the Goodman-Buckler panel. These analyses are now shown in a new **Supplementary Figure 14** (referenced on **line 411**). As shown in the new Supplementary Fig. 14A, our method clearly detected PAV. We confirmed these sequencing data with PCR using primers targeting the 3' region of the SV and the 5' UTR region of *ereb184*.

The association between the PAV and LA or TBN for this subset of 49 maize lines is reported in Supplementary Fig. 14B, and the P-values are reduced (Wilcoxon, $p = 0.02$ and $p = 0.04$ for LA and TBN, respectively).

- A few technical details: Why $r^2=0.7$ was used for LD filtering (line 187 in Method)? Why an FDR-adjusted P-value below 0.2 was used (line 224 in Method)? How expression time was calculated? More details should be provided.

Response: We removed single nucleotide polymorphisms (SNPs) that exhibited high linkage disequilibrium (LD), using an LD threshold of 0.7 within a 50-base-pair (bp) window. This approach was taken to reduce redundancy in the SNP dataset when estimating kinship. By pruning to remove markers in high LD (0.7), we retained a set of representative markers that maximizes the likelihood of tagging potential causal genetic variants while minimizing redundancy.

We acknowledge the inherent conservativeness of genome-wide association studies (GWAS) and are mindful of the risk of overlooking biologically significant associations due to smaller effect sizes falling below traditional thresholds of statistical significance. To address this, we applied an adjusted false discovery rate (FDR) of < 0.2 to filter markers for downstream analysis. This slightly less stringent cutoff was chosen intentionally, as our marker set was pre-selected based on gene networks, allowing us to capture a broader range of potential associations. In GWAS, the choice of FDR cutoff is typically left to the researcher as a tool for refining the list of genes for further exploration. By adopting a more permissive threshold, we aimed to retain markers that may exhibit subtle but potentially meaningful associations that could be missed under stricter criteria.

The expression time analysis was described in the submitted Methods of the original manuscript (now lines 572-576). *“Expression time (ET) was calculated using a smooth spline regression model 5 with the R function bs(). We fitted a b-spline (3-knot with three degrees of freedom) modeled on the first and second PC of the 500 most dynamically expressed genes across normal tassel development. Data points from mutant backgrounds were classified based on their location on the spline in relation to this model.”*

Response to Reviewer 3:

This manuscript provides an in-depth analysis of the genetic and genomic networks responsible for the pleiotropy between tassel branch number and leaf angle in maize. An increased leaf angle has been repeatedly selected in maize to optimize light capture in dense plantings, while the associated reduction in branch number is of less certain agronomic relevance. The authors explored transcriptional profiles for multiple stages of tassels and leaves across a panel of mutants that influence leaf angle and tassel branch number. This rich expression data set facilitated construction of a regulatory network for the common leaf angle/tassel branch number phenotype, and a set of gene co-expression networks. These networks were then used to select a subset of SNPs likely to be associated with leaf angle and tassel branch number for increased statistical power in a GWAS analysis. The GWAS identified *ereb184* and *zhd21* as likely regulators of the pleiotropy in a large diversity panel. Functional validation of *zhd21* and *zhd1*, a related TF also implicated in the leaf angle/branch number pleiotropy, was performed with mutator insertions which a clear branch number and leaf angle phenotype for *zhd21*, and a complex epistatic interaction between *zhd21* and *zhd1*. Functional validation of *ereb184* was performed through a naturally present promoter variant which localized to a likely regulatory region. Surprisingly, lines containing this variant had an increase in branch number and more upright leaves, reversing the typical pleiotropy of these linked phenotypes.

Overall, I found this an impressive study. Considering the complexity of the dataset and analyses, the manuscript is concise and well written. The figures nicely summarize the data. I was particularly impressed with the diverse complementary approaches that were used to home in on novel regulators of the branch number/leaf angle connection in maize. Ultimately, the approach proved effective as they were able to validate a role for two novel regulators of this phenotype including a natural regulatory variant that reverses the typical pleiotropy. This functional confirmation validates the extensive effort and expense of the network analysis when combined with a GWAS. In my view, this study provides a compelling model of how to expand our list of candidate genes and breeding targets for developmental traits that can be more broadly employed in maize and beyond.

I have a few mostly minor comments that I believe can easily be addressed in a minor revision.

1. The claims about evolutionary constraint, pleiotropy and developmental plasticity in lines 74-76 are interesting, but I imagine somewhat contested as well. Are there some specific citations to support these claims?

Response: Thank you for noting this and we agree. We added a few reviews as references here: Auge et al 2019 is a review describing effects of genetic pleiotropy on trait variation in flowering time, Hill et al. 2020 discusses molecular and evolutionary processes underlying gene expression variation, and Pavličev and Cheverud 2015 discusses context dependency and the evolution of pleiotropy. We also changed “*major cause of evolutionary constraint*” to “*significant cause of evolutionary constraint*” on **line 86**.

2. Is there some way to indicate which stage (1-5) was sampled for each of the replicates in Figure 1c? Not critical, but could make the figure a bit more informative.

Response: Thank you for this suggestion! We have updated **Figure 1c** to indicate immature tassel stages 1–5, and we agree this makes the figure more intuitive.

3. I believe previous work referred to ereb184 as ZmANT1, perhaps that is preferable in this case since there are so many ereb’s and the orthologous ANT1 has a well described function in Arabidopsis?

Response: We appreciate the suggestion here. However, the maize TFome nomenclature established by Yilmaz et al. (2009) <https://doi.org/10.1104/pp.108.128579> is widely recognized as a standard in the maize research community, including as standards on MaizeGDB, and we have chosen to adhere to this system in our manuscript. Using the designation 'ereb###' provides a clear and unambiguous reference to the gene in question.

4. is the direction of the phenotype (i.e. reduced TBN and LA) for zhd21-1 expected given any expression changes in other mutants your profiled? I would expect zhd21 levels to be reduced in at least some of the mutants (lg1, lg2, fun).

Response: Based on our expression data in the mutants, *zhd21* was down-regulated in developing leaf/ligule samples in several mutants with upright LA phenotypes (*lg2*, *fun*, and *bri1*) but up-regulated in *wab1R*, consistent with *zhd21-1* mutants displaying upright LA. We also found that *zhd21* was up-regulated in tassel primordia of *ra1* and *ra2* mutants, which make more tassel branches, consistent with a reduction in TBN in the *zhd21-1* mutant. While these results trend in a consistent direction, we acknowledge that the interconnected *zhd* regulatory subnetworks underlying these traits appear to be very complex and potentially redundant.

We appreciate the suggestion from the reviewer to look into the expression of *zhd21* in the mutants and we added a sentence to the Discussion **lines 462-465**: “*Notably, while zhd21 was down-regulated in lg2 developing leaf/ligule, there was no significant*

difference in its expression in *lg2* tassels, which make few to no tassel branches, potentially consistent with *zhds* regulating pleiotropic effects between these traits”.

We realized that this type of question may be common within the larger plant developmental community, so we decided to develop a **publicly accessible Shiny app**, available at <https://edoardobertolini.shinyapps.io/MAIZE-TBLAR>, to make it easy to explore our large dataset. This is now referenced in the Methods (**lines 1122-1123**). Below are screenshots of a table and graph views from our interactive app, where users can explore the differential expression of specific genes in the mutants we profiled.

AGPv4	Log2FC	FDR	Comparison	maizeGDB symbol
Zm00001d041780	-1.10262077	1.1E-05	bri1 - Shoot Apex 2	zhd21
Zm00001d041780	-0.71185125	0.01213	fun - Shoot Apex 1	zhd21
Zm00001d041780	-0.70535417	0.01915	lg2 - Shoot Apex 2	zhd21
Zm00001d041780	0.88520878	0.00032	wab1R - Shoot Apex 1	zhd21
Zm00001d041780	0.54844244	0.0467	ra2 - stage1	zhd21
Zm00001d041780	0.61307739	0.0138	ra1 - stage1	zhd21
Zm00001d041780	0.63785	0.00558	ra2 - stage2	zhd21

Differential expression (DE) data of *zhd21* across different comparisons in the profiled mutants and tissues. The table reports the Log₂ Fold Change (Log₂FC) of DE (mutant vs. normal, in the same developmental condition).

Expression levels (TPM) of *zhd21* (Zm00001d041780) across the profiled mutants in tassel stages.

Expression levels (TPM) of *zhd21* (Zm00001d041780) across the profiled mutants in shoot apices.

5. line 342-344 “Since we hypothesize some degree of functional redundancy among *zhd* genes, lack of phenotypes for the other alleles was not surprising.” This reasoning would apply to *zhd1*, but not to *zhd21-2* since the other allele of *zhd21* had a significant phenotype. Is there something about the location of the mu insertion in *zhd21-2*, or its effect on expression levels that explains the lack of phenotype in this allele?

Response: Yes! We addressed this point in a response to a similar question from Reviewer 2. Please see that response above and the associated **Supplementary Fig. 12**, which was added for reader clarity.

6. The interaction of *zhd1* and *zhd21* was unusual and intriguing. Is there some reason the double mutant (homozygous at both loci) was not included?

Response: We appreciate this comment. We did mention in the Methods section that “Phenotypic data were not collected on double mutant plants due to delayed maturation”. The double mutant plants between *zhd21* and *zhd1* have more complex phenotypes including very late and abnormal flowering, making it unclear on how to developmentally stage the plants for accurate phenotypic comparisons. We feel that a separate study on the genetic and mechanistic relations between these *zhds* would be warranted and it is outside the scope of this study to branch into an intensive developmental analysis of stacked *zhd* mutants. Co-author Strable is currently working on a more comprehensive study of the role of these *zhds* in maize development.

7. The nature of the structural variant in *ereb184* was not clearly described. Is this simply an indel? That seems to be the case, but “structural variant” could imply other rearrangements. Perhaps this could be shown graphically somehow in a figure or supplement?

Response: We appreciate this suggestion. In the Discussion at line 448 of the original manuscript (now lines 477-480) we did address this point by reporting the results of a blast search that revealed similarity with retrotransposons and Helitron elements. “*A blast search of the SV DNA sequence against a curated transposable element database revealed the presence of a probable LTR Gypsy retrotransposon (86% alignment identity) in the 3-prime region of the SV, as well as interspersed Helitron fragments*”.

To make this clearer for readers, we now include a graphical representation of the structural variant in the genome. To further validate our findings, we selected 49 inbred lines based on genetic similarity (kinship) with the NAM founders and plotted, as genome view, the sequencing coverage that we used to call the PAV of the SV across the Goodman-Buckler panel. This is now presented in **Supplementary Fig. 14 (line 411)**. We further confirmed this by PCR with primers targeting the 3' SV region and the 5' UTR region of *ereb184*.